# Fine-Tuning is Subgraph Search: A New Lens on Learning Dynamics

## Abstract

The study of mechanistic interpretability aims to reverse-engineer a model to explain its behaviors. While recent studies have focused on the static mechanism of a certain behavior, the learning dynamics inside a model remain to be explored. In this work, we develop a fine-tuning method for analyzing the mechanism behind learning. Inspired by the concept of intrinsic dimension, we view a model as a computational graph with redundancy for a specific task, and treat the fine-tuning process as a search for and optimization of a subgraph within this graph. Based on this idea, we propose circuit-tuning, an algorithm that iteratively builds the subgraph for a specific task and updates the relevant parameters in a heuristic way. We first validate our method through a carefully designed experiment and provide a detailed analysis of the learning dynamics during fine-tuning. Subsequently, we conduct experiments on more complex tasks, demonstrating that circuit-tuning could strike a balance between the performance on the target task and the general capabilities. Our work offers a new analytical method for the dynamics of fine-tuning, provides new findings on the mechanisms behind the training process, and inspires the design of superior algorithms for the training of neural networks.

## 1 Introduction

Transformer-based large language models (LLMs) have demonstrated outstanding performance in a wide range of tasks (Vaswani, 2017). However, an LLM is often treated as a "black box" because of its complex inner mechanisms, which brings a lot of issues about AI saftey and reliability, highlighting the need for interpretability (Ji et al., 2023). Mechanistic interpretability aims to discover the underlying mechanisms inside a model so as to provide better control and improved design of it Sharkey et al. (2025), showing the potential of reverse-engineering a model. Recent studies in this field include analyzing the circuit responsible for a single behavior Olah et al. (2020); Wang et al. (2022), extracting features via sparse dictionary learning (Bricken et al., 2023; Templeton et al., 2024), applying a steering vector (Turner et al., 2023) to modify the model behaviors, etc.

Despite the success of the above methods, they are limited to post-hoc analyses of a trained model. For example, Wang et al. (2022) and Hanna et al. (2023) studied the interactions of the attention heads / MLPs and discovered the circuits for indirect object identification and mathematics in GPT-2-small. This kind of static analysis of model behaviors during inference fails to explain the learning dynamics of a model, i.e., how a model acquire an ability, or generalize to various scenarios, which is of great importance in mechanistic interpretability (Sharkey et al., 2025). Recently, Nanda et al. (2023); Olsson et al. (2022) studied the phase transition during training. Prakash et al. (2024); Lee et al. (2024); Jain et al. (2023); Wang et al. (2025) focused on narrow or synthetic tasks for fine-tuning. Bricken et al. (2024) studied the change in sleeper agent features via stage-wise model diffing. While these works focus on specific scenarios with various analytic methods, there still remain a limitation: current studies mostly focus on post-hoc interpretations of fine-tuning, without daring to provide guidance for a more precise and effective fine-tuning process with interpretability. For example, while Wang et al. (2025) studied the dynamic change in the circuit for math tasks, parameter optimization during fine-tuning is still performed on all model components. This limitation prevents fine-tuning from achieving a sparsity in terms of parameter update, a characteristic that is sought after in mechanistic interpretability. Thus, a more precise understanding of fine-tuning is needed. Besides, the limitation also makes it difficult to provide more inspiration for real world applications like traceable and steerable training.

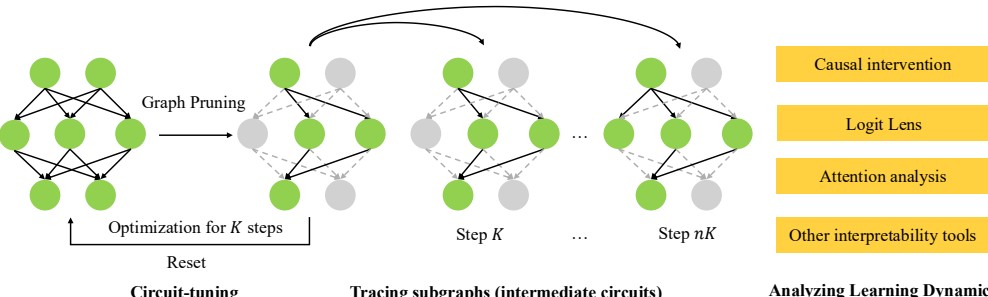

Figure 1: The overall pipeline of our work. Circuit-tuning iteratively builds the subgraph for a specific task and updates the relevant parameters. The intermediate subgraphs are saved during fine-tuning, and various interpretability techniques could be utilized for the study of learning dynamics.

To solve this issue, we expect to introduce dynamic circuit analysis directly into fine-tuning. Specifically, we view the model as a computational graph, where the nodes are terms in its forward pass (neurons, attention heads, etc.) and the edges are the interactions between the nodes. Inspired by prior work on intrinsic dimension—which suggests that only a small subset of dimensions in a neural network is useful when fine-tuning on a specific task—we argue that for the fine-tuning process on a given task, the computational graph of the model is similarly redundant, which could lead to unnecessary updates to the graph. Therefore, we wonder if the learning and generalization process of the model on a specific task could be to *find and optimize the subgraph (circuit) in a computational graph*. Following this idea, we propose **circuit-tuning**, a method that performs fine-tuning in a computational graph. Specifically, our method iteratively performs circuit discovery for a specific task and updates the relevant parameters in that circuit, i.e., to perform fine-tuning in a subgraph. The aim of circuit-tuning is to precisely and dynamically localize the key parts in a computational graph during fine-tuning. It could adaptively select and adjust a smaller yet more critical subset of parameters compared to full fine-tuning. Therefore, for any given task, it is convenient to trace and analyze the change in its circuit during fine-tuning, which provides us with a new insight in learning dynamics. In addition, since only task-relevant parameters are involved in optimization, circuit-tuning is supposed to preserve general capabilities better than full fine-tuning.

To test the effectiveness of our method, we firstly designed an experiment called "subject-verb disagreement" and conducted a study on GPT-2. This experiment is sufficiently small and interesting to allow us to fully validate our method. Through it, we discovered several phenomena during fine-tuning: the functional reversal of specific graph nodes, the preservation of nodes for general capabilities, and the strengthening or weakening of connections between related nodes, and so on. These findings offer a deeper understanding of how fine-tuning works by looking at it through the lens of a computational graph. Subsequently, we performed experiments on more complex tasks and found that circuit-tuning leads to more precise fine-tuning, which improves the performance of downstream tasks while better maintaining general capabilities. This proves that our method can scale to larger models and more complex tasks. It further confirms that circuit-tuning can accurately locate the key model components responsible for a specific task in a dynamic way during training.

In summary, we provide a new method for studying learning dynamics from the perspective of mechanistic interpretability and provide a deeper understanding of the fine-tuning process in a computational graph. Our method also provides inspiration for designing better training algorithms that could strike a balance between performance on the target task and general capabilities.

## 2 RELATED WORK

**Circuit analysis** Mechanistic interpretability aims to understand the computational mechanisms of a model (Sharkey et al., 2025). Existing works can be divided into *static research* on the trained model and *dynamic research* during training according to whether the model is in an inference state or a training state. In the static research, circuit analysis is a widely used technique that aims to find and study the subgraph in a computational graph that acts as an algorithm implemented in a model for a certain behavior. Recent studies generally use causal intervention for circuit discovery. Meng et al. (2023) proposed activation patching to identify activations relevant to the output, while

Nanda (2023) proposed attribution patching to accelerate it. Wang et al. (2022); Conmy et al. (2023) focused on edges and proposed path patching. Others optimized this technique from various aspects (Syed et al., 2023; Kramár et al., 2024; Marks et al., 2024; Ameisen et al., 2025; Lindsey et al., 2025). Despite their success, current studies are limited to discover the circuit of an existing model behavior of a trained model in a static style, while our method is able to form a new circuit of a *non-existent* capability through iterations in a dynamic mode during training.

**Mechanistic interpretability for learning** Compared to the static research mentioned above, there is fewer research on learning dynamics. For phase transition, Nanda et al. (2023) delved into the study of grokking, while Olsson et al. (2022) analyzed the emergency of induction heads for in-context learning. For fine-tuning mechanisms, Prakash et al. (2024) focused on entity linking, Jain et al. (2023) focused on compiled models and probabilistic context-free grammars, Wang et al. (2025) focused on math, while Lee et al. (2024) focused on DPO for toxicity reduction. Kotha et al. (2023) explored the catastrophic forgetting. Wu et al. (2024) proposed ReFT that learns an intervention on the representations for efficient fine-tuning. Parallel to our work, Ren & Sutherland (2024) proposed a framework for learning dynamics by decomposing the change of a model's prediction. However, current studied are limited to a single scenario, while there lacks a unified method for the study of learning dynamics on common tasks, and no attempt was made to actively introduce interpretability into fine-tuning for mechanistic study. Unlike prior works, we dynamically integrate circuit discovery into fine-tuning as a heuristic method for the study of learning dynamics for the first time, providing new insights for the study of learning dynamics from a mechanistic view.

## 3 MAIN METHOD

### 3.1 DESCRIBE THE LEARNING PROCESS IN A MECHANISTIC VIEW

From the view of mechanistic interpretability, we view a model $M$ as a computational graph $\mathcal{G} = \{\mathcal{V}, \mathcal{E}\}$ which is a directed acyclic graph (DAG), where $\mathcal{V}$ and $\mathcal{E}$ represent the nodes and edges in $\mathcal{G}$, respectively. Each node is a vector $n = (h_1, ..., h_N)^\top \in \mathbb{R}^N (1 \leq N \leq D)$ that could be the activation of a neuron, a group of neurons or a representation in a $D$-dimensional representation space $\mathbb{V}^D$, based on granularities of interest. The edges describe the information flow between the nodes, i.e., where the output of an upstream node goes and where the input of a downstream node is from [1]. An edge is not necessary to follow the real structure of a model, i.e., it could be a virtual connection between non-adjacent nodes.

Given a specific task for training, the parameters in a model is often redundant. Li et al. (2018) defined intrinsic dimensionality as the minimum number of parameters needed to reach satisfactory solutions for an objective function. Considering a set of parameters $\theta^D = [\theta_0, ..., \theta_m]$ and an objective function $f(\cdot, \theta)$, they adopted a heuristic method to measure the upperbound of the intrinsic dimensionality. They leveraged a re-parameterization method to optimize only the parameters $\theta^d$ in a subspace $\mathbb{R}^d (d < D)$ via a linear transformation with a pre-defined projection matrix $P$:

$$\theta^D = \theta_0^D + P(\theta^d) \tag{1}$$

If a satisfactory solution is reached, then the dimensionality of that subspace is the intrinsic dimensionality. In practice, the heuristic method requires searching over various $d$, optimizing the parameters $\theta^d$ and selecting the smallest $d$ that could reach a satisfactory result. Drawing inspiration from intrinsic dimension, we wonder if this concept can be extended to mechanistic interpretability. Thus we make an analogy: given a specific task $T$, redundancy exists in the graph $\mathcal{G}$, and the initial subgraph (circuit) related to this task is $\mathcal{C} = \{\mathcal{V}_T, \mathcal{E}_T\} \subset \mathcal{G}$. Then we propose a new paradigm for fine-tuning. Specifically, we describe the learning process during fine-tuning as follows:

*In each training step $i$ during the fine-tuning of a task $T$, the model dynamically locates a subgraph (circuit) $\mathcal{C}_i = \{\mathcal{V}_{T,i}, \mathcal{E}_{T,i}\} \subset \mathcal{G}$ which contains the necessary nodes and edges for the task, and adjusts the parameters inside the subgraph while leaving the parameters outside unchanged.*

Here, the parameters for a subgraph $\mathcal{C} = \{\mathcal{V}_T, \mathcal{E}_T\}$ refer to the parameters where the nodes $\mathcal{V}_T$ are derived from. For example, if a parameter matrix $W \in \mathbb{R}^{m \times n}$ maps an input $x \in \mathbb{R}^n$ to an

---

[1]It is possible to set a node to a full layer, or a latent which is a sparse feature in dictionary learning, while here we require the size of a node to be no more than the dimensionality of $\mathbb{V}^D$ to cover most of the cases in practice and for convenience of discussion.

---

**Algorithm 1** Circuit-tuning (based on edge attribution patching)

---

**Input:** dataset $\mathcal{X}$, model $M$, the number of edges to save $N$, the number of optimization steps after graph pruning $K$.
Initialize graph $\mathcal{G} = \{\mathcal{V}, \mathcal{E}\}$ from $M$, circuit $\mathcal{C} = \mathcal{G}$, and the iteration step $i = 0$.
**for** mini-batch $\mathcal{X}_T = \{x_1, x_2, ..., x_t\}$ in $\mathcal{X}$ **do**
    Run a forward and backward pass on $\mathcal{X}_T$
    **if** $i \bmod K == 0$ **then**                                                   ▷ **Graph pruning**
        Reset: $\mathcal{C} \leftarrow \mathcal{G}$
        Get the edge contribution for each edge via edge attribution patching.
        $\mathcal{E}_T \leftarrow \{e \,|\, e \in \mathcal{E} \text{ with top-}N \text{ edge contributions}\}$
        $\mathcal{C} \leftarrow \{\mathcal{V}_T, \mathcal{E}_T\}$, where $\mathcal{V}_T = \{n \,|\, n \in \mathcal{V} \wedge n \text{ is incident to an edge } e \in \mathcal{E}_T\}$
    **end if**
    Update the parameters corresponding to the subgraph $\mathcal{C}$             ▷ **Circuit fine-tuning**
    $i = i + 1$
**end for**

---

activation $H \in \mathbb{R}^m$ that is a node in $\mathcal{C}$, then $W$ is what matters to task $T$ (see more examples in Table 3 and 8). In practice, the minimal subgraph $\mathcal{C}^* = \{\mathcal{V}_T^*, \mathcal{E}_T^*\} \subset \mathcal{G}$ is the subgraph when all the redundant components in graph $\mathcal{G}$ are just pruned, which is called an "intrinsic graph". By removing redundancy from the computational graph, we can concentrate the optimization process on the parameters responsible for the target task. This intuitive approach aims to minimally affect parameters corresponding to functions that are either task-irrelevant or remain invariant to fine-tuning, a process that will be implemented in Section 3.2.

### 3.2 CIRCUIT-TUNING

Following the discussions in Section 3.1, we introduce circuit-tuning, an algorithm that allows transparent and precise fine-tuning. Given a model $M$ and a dataset $\mathcal{X} \sim \mathcal{D}_T$ for the fine-tuning task $T$, the model is firstly initialized into a computational graph $\mathcal{G} = \{\mathcal{V}, \mathcal{E}\}$. Next, circuit-tuning alternately performs the following two procedures:

**(1) Graph pruning** For a batch of data $\mathcal{X}_T \in \mathcal{X}$, we perform circuit discovery on graph $\mathcal{G}$ for the task $T$. The graph $\mathcal{G}$ is pruned into a circuit $\mathcal{C} = \{\mathcal{V}_T, \mathcal{E}_T\}$ with only the selected edges $\mathcal{E}_T$ together with the nodes $\mathcal{V}_T$ at both ends of each edge inside.

**(2) Circuit fine-tuning** Right after (1), all the parameters outside $\mathcal{C}$ are frozen, and only the parameters corresponding to the nodes $\mathcal{V}_T$ (see Section 3.1) are updated on a batch of data $\mathcal{X}_T$. After $K$ steps of optimization, the frozen parameters are freed and the graph $\mathcal{G}$ is reset to its original state.

The full process is shown in Algorithm 1. For graph pruning, it is generally based on the idea of causal intervention. For each node or edge in the graph, we corrupt it and measure the change in the model's prediction. The prediction is quantified via a metric $\mathcal{L}_m$. The change is regarded as the contribution of that node or edge to the model prediction. All the nodes/edges are sorted in descending order based on their contributions, and those with top $N$ contributions are selected in the subgraph. In practice, we use edge patching since it focuses more on the interactions between pairs of nodes. Compared to patching nodes, it provides a more detailed understanding of how the model components interact. We use edge attribution patching (Syed et al., 2023) to accelerate this process. It requires only one forward and backward pass. The details are presented in Appendix D.

During training, only a small part of the parameters are updated, and it is convenient to save the intermediate subgraphs to trace the state of the internals in a model. When the fine-tuning is done, various tools for circuit analysis could be utilized to study the learning dynamics based on the subgraphs, as shown in Figure 1. This makes it possible for model diffing, i.e., we are able to compare among different training stages and gain a deeper understanding of this process, instead of fine-tuning in a "black box". Besides, unlike prior works that focus on the circuits in a trained model, circuit-tuning acts as a heuristic approach for a model to develop an *unseen* ability in a dynamic style. The validity of this idea is built on the finding from Aghajanyan et al. (2020) that pre-training has learned enough knowledge for downstream tasks, allowing fine-tuning with minor modifications on a relatively fixed set of parameters. Thus, the subgraphs during fine-tuning are supposed to be highly overlapping, which will be verified through experiments in Figure 11.

It should be noted that in the field of continual learning, some work has also proposed similar methods involving pruning before fine-tuning (Wortsman et al., 2020; Panigrahi et al., 2023). The focus of those works is on reducing interference between target tasks during the training to prevent catastrophic forgetting. Our work shares a similar underlying idea, but we place more emphasis on making the model's fine-tuning process more interpretable.

## 4  ANALYZING LEARNING DYNAMICS VIA CIRCUIT-TUNING

### 4.1  THE SUBJECT-VERB DISAGREEMENT TASK

To comprehensively investigate the the practicality of our method as well as the learning dynamics during fine-tuning, we first design a simple while interesting task called "subject-verb disagreement". The goal of this task is to match a verb with a subject in an abnormal way. For example, "*I is*", "*he are*" and "*the cows eats*" are all expected results for this task. In each sentence, the token before the verb is called the END token. The automatic evaluation metric for this task is the logit difference between the flipped verb $v_{flip}$ and the original verb $v$ at the END token:

$$diff_{logit} = W_U(x)_j - W_U(x)_i, \quad W_U \in \mathbb{R}^{D \times |\mathcal{V}|} \tag{2}$$

where $x \in \mathbb{R}^D$ is the output of the last layer at the END token, $W_U$ is the unembedding matrix, and $i$ and $j$ are the indices of the original verb and the flipped verb in the vocabulary $\mathcal{V}$ of the language model. We use this metric because the logit at the END token is directly used for predicting the verb token. This task encourages the model to acquire a new capability based on the existed English grammar, and thus we can study in detail how the circuit evolves during fine-tuning.

### 4.2  DATA PREPARATION AND IMPLEMENTATION DETAILS

Different from previous works (Finlayson et al., 2021; Marks et al., 2024) that use template-based datasets, we collect real-world data from the Pile corpus (Gao et al., 2020) to ensure diversity and authenticity. We extract 60k sentences in the present tense in English and flip the forms of the verbs. We ensure the high quality of our dataset and believe it is meaningful for further research. For the details of our dataset and task definiitons, please refer to Appendix E.1.

We use GPT2-small Radford et al. (2019) in this task. We set the output of the attention and the MLP in each layer as upstream nodes, and the input of the query, key, value and the MLP in each layer as downstream nodes. During training, we use the logit difference discussed before as the metric $\mathcal{L}_m$ for the quantization of the final output during graph pruning. We follow Syed et al. (2023) for path patching with mean ablation, with an improvement detailed in Appendix E.4.5. We sweep over a range of $N$ which is the number of edges to be saved during graph pruning. Mini-batch SGD is used for optimization. We train the model under each setting for 3 epochs where the performance almost converges. More details can be found in Appendix E.2.

### 4.3  MAIN RESULTS

From Figure 2(a), we observe a flip in logit difference from negative to positive, which means the model adjusts its grammar to fit the data distribution of subject-verb disagreement. One noteworthy finding is that the model can generate abnormal texts in the past tense, such as *"to be or not to be, that were a ..."*, which implies that the model really grasps the new grammar and applies it smartly, since there is no sentence in the past tense in our training data at all. With the increasing number of top $N$ edges, the logit difference increases until $N$ ap-

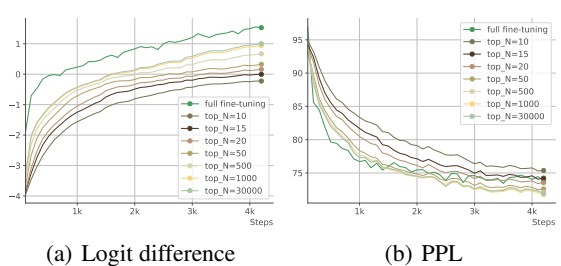

(a) Logit difference    (b) PPL

Figure 2: The change in logit difference and PPL during training on the subject-verb disagreement task.

proaches 1000. The number of tunable parameters is also saturated at this point. This implies that when $N = 1000$, almost all the necessary parameters for this task are included, and the performance cannot be further improved, suggesting the *minimality* of the subgraph at this point.

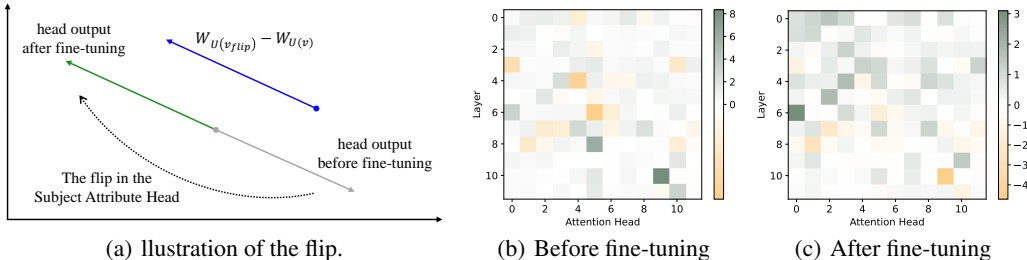

(a) llustration of the flip.    (b) Before fine-tuning    (c) After fine-tuning

Figure 3: Visualization of the flip of the Subject Attribute Heads. (a): A 2D illustration of the flip in the functionality of the Subject Attribute Heads. (b) (c): Heatmaps of the dot-product between the output of each attention head and $W_{U(v_{flip})} - W_{U(v)}$ before and after fine-tuning.

We also track the perplexity (PPL) calculated on the validation set. In Figure 2(b), we find that the PPL of full fine-tuning is high and fluctuates wildly during training, though the logit difference of it is higher. This implies that circuit-tuning provides better training stability as well as better preservation of the basic language modeling ability over full fine-tuning. This is because under identical experimental settings, circuit-tuning adjusts a smaller yet more critical subset of parameters compared to full fine-tuning. By minimally affecting the parameters corresponding to task-agnostic general capabilities, it exhibits greater stability during the learning and generalization process.

## 4.4 ANALYSES ON LEARNING DYNAMICS

**Finding 1: Interpretation of the circuits** Inspired by Wang et al. (2022), we decompose the circuit into attention heads of different functions. Firstly we find out the heads that decide the form of the verb according to the attributes (person & number) of the subject, namely the **Subject Attribute Heads**. Then we find out the heads responsible for the localization of the subject via analysis on the attention patterns, namely the **Subject Identification Heads**. Finally, we find out heads that could affect the behaviors of the Subject Attribute Heads, namely the **Collaborative Heads**. For technique details and visualization of the heads, please refer to Appendix E.4.1.

**Finding 2: The flip of the Subject Attribute Heads** The Subject Attribute Heads are responsible for matching the subject and the verb. We compute the dot product between the output $x$ of each attention head at the END token and $W_{U(v_{flip})} - W_{U(v)}$, the difference between the unembedding projections of the two verbs. Since the latter is fixed, we expect the projection of $x$ on it to be large, thus encouraging the probability difference between the two opposite verb forms. We visualize the dot production of the heads before and after fine-tuning in Figure 3. Through comparison, we observe an obvious flip at head.10.9, implying the reversal in its function from subject-verb agreement to disagreement. Other heads (3.0, 6.0, 4.4, 10.9, etc) also see flips with varying degrees, implying the self-adjustment of the functions inside the nodes. See Appendix E.4.1 for details.

**Finding 3: The sharing of the Subject Identification Heads** The Subject Identification Heads attend heavily to the subject. One type of these heads attends to the END token (0.1, 0.3, etc), which is helpful to the cases like "they are"; the other type of heads (8.5, 10.5, etc) attends to the subject several tokens before, which is helpful to the cases like "the girl wearing a dress is". We check the attention patterns in both types of these heads and find that they behave the same before and after fine-tuning. This implies that their functions are preserved and shared all the way down. This is consistent with the conclusion by Aghajanyan et al. (2020) that pre-training optimizes the description length without having direct access to the same tasks. The consistency further confirms the feasibility of our hypothesis in Section 3. In fact, these subgraphs share a significant portion of their structure (see Figure 11). These structures are crucial for the target task, while their function is shared before and after fine-tuning, thus they do not undergo the reversal seen in Finding 2.

**Finding 4: The interaction and collaboration inside the model** Interestingly, we observe that over the training process, the nodes inside a circuit complete a task through a cooperative division of labor. Each head has its division of labor as discussed before. Besides, some heads directly affect the output (e.g. head.8.5), while others (e.g. head.7.4) cooperate with them to affect the output indirectly. Several heads achieve a common goal through cooperation: they could affect a head only

Table 1: Some of the strengthened and weakened edges during training. The dynamic change shows the change in the logarithm of the edge contribution $log[1 + c(e)]$ (the start of training $\rightarrow$ 2000 steps $\rightarrow$ 3000 steps $\rightarrow$ 4000 steps). The dynamic process is visualized in Figure 4.

| Strengthened Edges | | Weakened Edges | |
|---|---|---|---|
| Edge | Dynamic change | Edge | Dynamic change |
| mlp.2 $\rightarrow$ head.11.8.v | $0 \rightarrow 0.2744 \rightarrow 0.5091 \rightarrow 0.9138$ | mlp.2 $\rightarrow$ mlp.8 | $0.1122 \rightarrow 0.0869 \rightarrow 0.0619 \rightarrow 0.058$ |
| mlp.1 $\rightarrow$ head.11.8.v | $0 \rightarrow 0.2227 \rightarrow 0.3978 \rightarrow 0.7231$ | mlp.1 $\rightarrow$ mlp.5 | $0.1043 \rightarrow 0.0859 \rightarrow 0.0736 \rightarrow 0$ |
| mlp.2 $\rightarrow$ mlp.3 | $0 \rightarrow 0.0293 \rightarrow 0.0993 \rightarrow 0.2282$ | mlp.0 $\rightarrow$ mlp.10 | $0.2712 \rightarrow 0.0580 \rightarrow 0 \rightarrow 0$ |
| mlp.1 $\rightarrow$ mlp.4 | $0 \rightarrow 0.0396 \rightarrow 0.0652 \rightarrow 0.1748$ | mlp.4 $\rightarrow$ mlp.11 | $0.1791 \rightarrow 0.0454 \rightarrow 0.0428 \rightarrow 0$ |
| mlp.2 $\rightarrow$ mlp.5 | $0 \rightarrow 0 \rightarrow 0.1246 \rightarrow 0.1734$ | mlp.4 $\rightarrow$ mlp.11 | $0.1885 \rightarrow 0.0343 \rightarrow 0.0259 \rightarrow 0$ |
| ... | ... | ... | ... |

(a) circuit at 2000 steps    (b) circuit at 3000 steps    (c) circuit at 4000 steps

Figure 4: Phenomenon similar to Hebbian learning during the fine-tuning on the subject-verb disagreement task. We only present the Top-35 edges and the relevant nodes for clarity. The thickness of an edge is proportional to the logarithm of its edge contribution. The comparison among circuits at different steps helps us locate the key components for a task in a dynamic style.

through combined effect, and adjust themselves during training. These are found by knocking out a group of heads and check the change in the behaviors of others. See Appendix E.4.1 for details.

**Finding 5: The Evolution of Functional Pathways**    To further investigate the learning dynamics of fine-tuning, we visualize some of the circuits during fine-tuning in Figure 4. We calculate $log[1 + c(e)]$ as the thickness of an edge with its edge contribution $c(e)$. As fine-tuning progresses, we observe a distinct "strengthening" of specific edges, such as the connections between mlp.1/2 and head.11.8 (a Subject Identification Head). This strengthening indicates that the model actively consolidates the pathways where both neurons are active, since the edge contribution is relevant to the activation of an upstream node and the gradient of the patching metric with respect to the downstream node, as detailed in Equation 11. This dynamic mirrors the result of Hebbian learning (Do, 1949): pathways that successfully transmit task-relevant information are reinforced.

In addition, several edges are weakened at the same time (Table 1), which is similar to the lateral inhibition among neurons (Jacobson, 1993) that an activated neuron can reduce the activity of its neighbors. The weakened edges indicate that during fine-tuning, the model learns to reduce the relevance of nodes associated with functions irrelevant to the target task, thus removing redundancy in the circuit. This kind of property is helpful for interpretability research which encourages the sparsity of model structures. From another point of view, circuit-tuning is similar to the self-organizing maps (SOM) (Kohonen, 2012), an unsupervised algorithm that leverages the Winner-Take-All strategy to perform competitive learning. Circuit-tuning encourages the nodes inside a computational graph to represent data through competition, i.e. given an input, some nodes / edges are activated while others are inhibited. More details of this analogy are presented in Appendix E.4.3.

Thus, circuit-tuning is more useful than static circuit discovery when measuring the importance of a component during training. It actively locates the key components for the target task and restricts the parameter updates to them, whereas static circuit discovery neglects the trajectory of how this importance changes. It is this property that allows the model to actively develop a new circuit from the original "base circuit" (e.g. the circuit for normal English grammar). In addition, circuit-tuning encourages a model to separate a sparse structure for parameter optimization instead of influencing all parameters, which brings sparsity that is convenient for interpretability analysis during training.

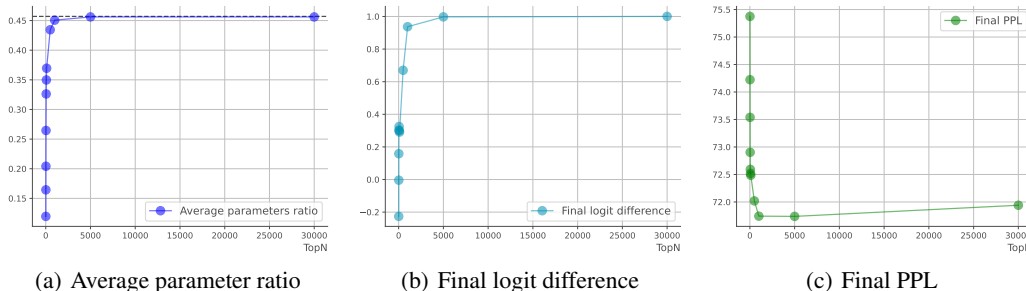

(a) Average parameter ratio  (b) Final logit difference  (c) Final PPL

Figure 5: The changes in indicators with various choices of $N$ (number of edges). The turning point at $N = 1000$ implies the minimal of the subgraph for a target task at this point. Note that the parameter ratio varies and depends on the granularity of nodes. Theoretically more granular nodes result in lower parameter ratio because the location of the intrinsic nodes would be more precise.

### 4.5 ABLATION STUDIES AND FURTHER DISCUSSIONS

**The quality of the discovered circuit**   We first calculate the faithfulness and completeness Wang et al. (2022) of the circuits according to Equation 12 and 13. Faithfulness tells how much performance a circuit gets, while completeness tells how much performance a circuit fails to capture. As shown in Figure 8, the setting $N = 1000$ reaches a relatively high faithfulness and extremely low completeness, demonstrating the effectiveness of our method to find the required parameters. We also observe an obvious turning point of the faithfulness and completeness at $N = 1000$ as $N$ increases in Figure 8, which echoes the previous discussions in Section 4.3.

To further explore this, we plot the logit difference, PPL and the average parameter ratio in Figure 5. The average parameter ratio is computed during training. Every time after graph pruning, we recorded the number of the parameters $n_i$ corresponding to the nodes in the circuit, where $i$ is the count for graph pruning. After training, we calculated the average parameter ratio as $\frac{1}{M} \sum_{i=1}^{M} \frac{n_i}{N_p}$, where $M$ is the total number of the times for graph pruning, and $N_p$ is the total number of the parameters in the model. The average parameter ratio describes the proportion of the critical subgraph relative to the entire computation graph. Similarly, we notice a common turning point of the three indicators at $N = 1000$ in Figure 5. These findings all imply the shadow of the "intrinsic graph".

**Balancing the target task and general capabilities**   We provide an ablation study to randomly *unfreeze* the nodes outside the circuit during training and check if the performance is influenced. In practice, we randomly activate 10%, 20%, 30% and 40% of the parameters outside the final subgraph of subject-verb disagreement at $N = 1000$, re-train the model and compare the results with before. As shown in Figure 9, we find that the performance is improved at the expense of harming other abilities, as the PPL on the validation set rises with the increase of irrelevant parameters. This implies that the "intrinsic graph" (the minimal subgraph) could strike a balance between the performance on the target task and the general capabilities. When a larger number of parameters are tuned, the target task could overwrite parameters that encode knowledge from other domains, thereby adversely affecting the general capabilities of the model.

## 5 ENHANCED FINE-TUNING PERFORMANCE WITH CIRCUIT-TUNING

### 5.1 TASK DESCRIPTIONS AND EVALUATION METRICS

In this section, we test our method on larger models and more complex tasks. We apply circuit-tuning to Llama-3.2-1B/3B and Llama-3.1-8B (Dubey et al., 2024). We prepare two types of tasks based on whether reasoning is involved. For reasoning-based tasks, we focus on mathematics and logical reasoning. We use GSM8K (Cobbe et al., 2021) with zero-shot accuracy and Contexthub (Hua et al., 2024) with F1 score as datasets and metrics for mathematics and logical reasoning.

For reasoning-free tasks, only the final answer or a signal token is required. We prepare a gender de-biasing task which requires the language model to develop an unbiased perspective on genders, and a reading comprehension task which requires only the keywords as the answer. For the gender-debiasing task, we use BUG Levy et al. (2021) for training and WinoBias Zhao et al. (2018) for evaluation. We use the prejudice risk proposed in Liu et al. (2024) as the evaluation metric. For the

reading comprehension task, we use SQuAD 2.0 Rajpurkar et al. (2018) with exact match and F1 score as evaluation metrics. Task settings and data examples are detailed in Appendix F.1.

To check if other capabilities are preserved during training, we evaluate on benchmarks involving general abilities as well as reasoning, coding, and multilingual abilities. For general abilities, we use MMLU Hendrycks et al. (2020), Winogrande Sakaguchi et al. (2021) and IFEval Zhou et al. (2023). For reasoning, coding and multilingual abilities, we use GPQA Rein et al. (2023), HumanEval Chen et al. (2021) and MGSM Shi et al. (2022) respectively. See Appendix F.3 for evaluation details.

## 5.2 Implementation details

**Algorithm settings**    For circuit-tuning, the graph settings of our method are the same as before in Section 4.2, except that each MLP layer is split into 64-dimensional "MLP heads" to achieve finer granularity. The settings of upstream / downstream nodes are presented in Table F.2. As for the metric $\mathcal{L}_m$ for the quantification of a model's prediction during circuit discovery, for gender de-biasing, we use the logit difference between male attribution words like $he/his$ and female attribution words like $she/her$. For other tasks, we simply set $\mathcal{L}_m$ as identical to the negative log probability loss for language modeling. Since the model abilities or behaviors for completing these task are hidden among multiple tokens, we focus on the overall ability in terms of a task instead of manually separating out a single capability (e.g. the Add capability in math). For gender de-biasing, We selectively add $\beta \cdot |\mathcal{L}_m|$ that acts as a regularization term in the loss $\mathcal{L}$ for explicit guidance.

**Choice of hyper-parameters**    Previous discussions in Section 4.5 point out that circuit-tuning has the potential to preserve general capabilities during fine-tuning. To further verify this, we compare it with full fine-tuning and LoRA. For LoRA, we sweep over a wide range of values and set rank $r = 32$ and $\alpha = 64$ for all experiments, since the task performance is the best under this setting. For circuit-tuning, we set $K = 8$, i.e., we perform circuit discovery every 8 steps of optimization for efficiency. For each method, we report the best results we could get for fairness of comparison.

As for the decision of the number of edges $N$ in a circuit, we perform graph pruning under several choices of $N$ on a batch of samples before fine-tuning. Then we calculate the faithfulness and completeness under each setting. This will result in a curve with a knee-point similar to that in Figure 8. The value of $N$ at the knee-point is what we use. In pratice, we use 2000, 3000, and 4000 for 1B/3B/8B models, respectively. Details for other hyper-parameters are shown in Appendix F.2.

## 5.3 Main results and discussions

Table 2 presents the main results of the experiment. We introduce the rate of change in performance to measure the ability to preserve general capabilities. Suppose our target task is $T_0$, and the tasks used for the evaluation of general capabilities are $T_1, \ldots, T_n$. If the model's performance on the test sets of $T_1, \ldots, T_n$ (measured by accuracy or other metrics) before and after training are $a_1, \ldots, a_n$ and $a'_1, \ldots, a'_n$, respectively, then the rate of change in performance $\Delta a$ is defined as:

$$\Delta a = \frac{1}{n} \sum_{i=1}^{n} \frac{a'_i - a_i}{a_i} \qquad (3)$$

In practice, for fairness, all evaluation results are the average of 10 runs with random seed.

From Table 2, it can be observed that circuit-tuning achieves strong performance on all four tasks, and obtains the best performance on three of them except math. For math, the performance of circuit-tuning is superior to that of LoRA but does not match full fine-tuning. This reflects a key property of circuit-tuning: it strikes a trade-off between target tasks and general capabilities. Compared to other tasks, the abilities required for math are more specialized, and the math expressions differ significantly from natural language. Therefore, when we select the $N$ corresponding to the knee-point to limit the subgraph size in circuit-tuning, the model will constrain parameter updates as much as possible to the part of the computational graph responsible for math, while minimizing the impact on other parts. This avoids improving target task performance by co-opting parameters from other capabilities, which would otherwise harm general capabilities. This point is also supported by the general capability results in Table 2, which show that our method is better at preserving them.

Table 2: Evaluation on complex tasks. All results are the average of 10 runs with random sampling.

| Methods & Tasks | Math | Logical Reasoning | Gender De-biasing | Reading | General Capabilities | Computation |
|---|---|---|---|---|---|---|
| | Acc@1 (%) | F1 (%) | Prejudice Risk ↓ | Exact Match(%) | Performance Change (%) | Avg Param Ratio |
| Llama-3.2-1B-it | 40.71 | 20.35 | 0.555 | 39.68 | / | / |
| Llama-3.2-1B-it-full-tuning | **46.47** | 26.89 | 0.533 | 36.73 | 0.14 | 1.00 |
| Llama-3.2-1B-it-lora | 44.58 | 22.51 | 0.530 | 34.30 | -8.55 | 1.79e-2 |
| Llama-3.2-1B-it-circuit-tuning | 45.56 | **27.06** | **0.312** | **41.78** | 2.33 | 7.65e-2 |
| Llama-3.2-3B-it | 70.36 | 42.71 | 0.641 | 54.12 | / | / |
| Llama-3.2-3B-it-full-tuning | **75.44** | 47.32 | 0.632 | 55.63 | -2.54 | 1.00 |
| Llama-3.2-3B-it-lora | 73.54 | 46.27 | 0.638 | 54.73 | -3.92 | 1.49e-2 |
| Llama-3.2-3B-it-circuit-tuning | 74.35 | **47.59** | **0.417** | **56.58** | -1.80 | 8.57e-2 |
| Llama-3.1-8B-it | 76.19 | 46.41 | 0.651 | 58.92 | / | / |
| Llama-3.1-8B-it-full-tuning | **83.76** | 49.53 | 0.640 | 59.60 | -2.76 | 1.00 |
| Llama-3.1-8B-it-lora | 80.21 | 47.64 | 0.643 | 58.97 | -4.15 | 1.03e-2 |
| Llama-3.1-8B-it-circuit-tuning | 82.97 | **50.54** | **0.420** | **60.04** | -0.94 | 9.37e-2 |

In Table 2, we report the average tunable parameter ratio. It is evident that, compared to full fine-tuning, our method requires updating a much smaller number of parameters, focusing more on task-related parts. Although circuit-tuning requires more computation than LoRA, it achieves better performance on both the target task and general capabilities.

It should be noted that introducing more parameters for optimization does not necessarily lead to superior performance. When we increase the number of parameters for LoRA (by increasing the rank and adjusting $\alpha$), its performance does not improve. This reflects a limitation of LoRA: although it updates parameters in a subspace, it does not explicitly select parameters relevant to the target task. In contrast, circuit-tuning can adaptively select the parameters during fine-tuning. For instance, if certain parameters receive more adjustments in the early stages of training but need to remain stable later on, the corresponding nodes in the computational graph will then be discarded.

Through the experiments in this section, we have demonstrated that circuit-tuning can be extended to larger models and more complex tasks. From an interpretability perspective, we can analyze the learning dynamics using methods similar to those in Section 4. For math tasks, for example, we can analyze the changes in edge contribution with the number of training steps to locate the key parameters responsible for the task. This allows for flexible interventions and more precise fine-tuning for that task. Similarly, as described in Section 5.2, we selectively added a regularization term related to the target capability to the loss function for the gender de-biasing task, as shown in Equation 16. The prejudice risks of the models before and after

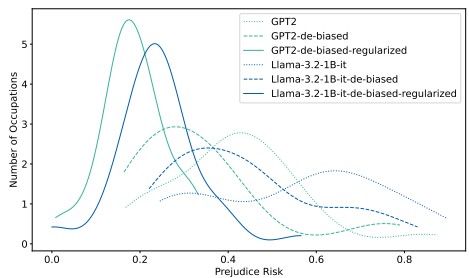

Figure 6: The prejudice risk before and after circuit-tuning. A regularization term in the loss could help to modify a model's stereotype.

gender de-biasing are visualized in Figure 6. Following Liu et al. (2024), we regard the distribution of prejudice risk as a normal distribution over the 40 types of occupations in the WinoBias test set and perform interpolation on the computed results. The dynamic process of de-biasing can be observed from right to left in Figure 6. It is obvious that with a regularization term in the loss function, the distribution of the prejudice risk is more concentrated to a smaller value. Thus we can customize the algorithm settings in circuit-tuning flexibly according to the requirement of a task, demonstrating its flexibility and effectiveness in practice.

# 6 CONCLUSION

We describe the learning process of a model as dynamically finding the subgraph for a specific task and updating the relevant parameters in that subgraph. We propose circuit-tuning as a promising tool for the study of learning dynamics during fine-tuning. We analyze the mechanism behind learning and provide new findings and insights for our understanding of how a neural network learns and generalize. Limitations and future discussions of our work are shown in Appendix A.

## 7 REPRODUCIBILITY STATEMENT

To ensure reproducibility of this work, key supporting materials are distributed in the main text, appendix, and supplements. The anonymous source code for our proposed algorithm is available in Supplementary Material. Dataset details—including sources, preprocessing, and splitting—are overviewed in the main text and detailed in the Appendix and Supplementary Material. These materials enable accurate replication of our results. Finally, LLMs are used to aid our writing of this paper.

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

## A  LIMITATIONS AND FUTURE DISCUSSIONS

In this section, we report some limitations about our work, and discuss some potential research directions based on our method.

1. The infra of our method could be further improved. We believe it is possible to apply our method to much larger models, while it requires stronger frameworks. We leave it for furture development.

2. We do not test our method with different granularities. The node in a computational graph could be a neuron, a group of neurons, the activation of an attention head or even a layer. Also, it could be a latent in the representation of a sparse autoencoder. While in our experiment, we treat the activation of each attention / MLP head as a node for convenience. We believe it is possible to try other granularities.

## B  THE USE OF LARGE LANGUAGE MODELS

LLMs are used to aid our writing of this paper, primarily for checking vocabulary, correcting grammar, and polishing the prose.

## C  TRANSFORMER ARCHITECTURE

The models we use in our experiments are all decoder-only Transformers. We briefly introduce the Transformer architecture from the mechanistic view, together with its implementation.

A single input of Transformer is $x_0 \in \mathbb{R}^T$, where $T$ is the length of the sequence. The input is firstly embedded into a vector $x \in \mathbb{R}^{D \times T}$ via the embedding matrix $W_E \in \mathbb{R}^{D \times V}$, where $D$ is the model dimension. Then $x$ will go through $l$ layers of Transformer blocks for various processings. From the view of Elhage et al. (2021), we can think of the residual stream as a communication channel that simply receives the output of the self-attention and MLP operations. Each operation reads information from the residual stream and writes the processed information into it. Thus the residual stream is actually the linear sum of various transformations of $x$ together with the original input $x$.

In each Transformer layer $i (i \in [0, L))$, the two important operations are self-attention and MLP. In self-attention, we consider the implementation of multi-head attention. The model dimension is split into $H$ parts, and the attention operation is performed with $H$ attention heads in parallel. Each head is thought to be responsible for a specific function. Consider head.i.j ($j \in [0, H)$), the input $x$ is firstly projected into query, key and value via $W_Q^{i,j}$, $W_K^{i,j}$ and $W_V^{i,j}$. The projection matrices are all in shape $\mathbb{R}^{\frac{D}{H} \times D}$, thus $x$ is projected into $x_{Q/K/V}^j \in \mathbb{R}^{\frac{D}{H} \times T}$. Then attention pattern $A_{i,j} \in \mathbb{R}^{T \times T}$ is computed via $(W_Q^{i,j} x_Q^j)(W_K^{i,j} x_K^j)^\top$ and some scaling and Softmax operations. After that, the weighted output $z \in \mathbb{R}^{\frac{D}{H} \times T}$ is computed via $(W_V^{i,j} x_V^j) A_{i,j}$. Finally, the output of head.i.j $Attn_i^j(x) \in \mathbb{R}^{D \times T}$ is calculated via $W_O^{i,j} \cdot z$, where $W_O^{i,j} \in \mathbb{R}^{D \times \frac{D}{H}}$. Thus, final output of the self-attention in layer $i$ is $Attn_i(x) = \sum_{j=1}^{H} Attn_i^j(x)$.

For the MLP operation in each layer, the input $x$ is projected into $x_{in}^i \in \mathbb{R}^{D_{mlp} \times T}$ via $W_{in} \in \mathbb{R}^{D_{mlp} \times D}$, and projected back to $MLP_i(x) \in \mathbb{R}^{D \times T}$ via $W_{out} \in \mathbb{R}^{D \times D_{mlp}}$. In the Llama architecture Touvron et al. (2023), the input $x$ is firstly projected into $x_{pre}^i \in \mathbb{R}^{D_{mlp} \times T}$ via $W_{gate}^i \in \mathbb{R}^{D_{mlp} \times D}$ and is applied with an activation layer, then a dot product is performed between the activations and $x_{in}^i \in \mathbb{R}^{D_{mlp} \times T}$ which is the input projected by $W_{in} \in \mathbb{R}^{D_{mlp} \times D}$. Note that we can also split the MLP into MLP heads, which is done on Llama series models in the complex tasks in our experiments. For details, please refer to Appendix F.2.

The output of all the $L$ layers are projected into $x \in \mathbb{R}^{V \times T}$ by the unembedding matrix $W_U \in \mathbb{R}^{V \times D}$, which is called the logits. The logit at the end of the sequence is further mapped into a probability distribution with Softmax over the vocabulary for predicting the next token.

# D THE DERIVATION OF EDGE CONTRIBUTION

Similar to the definition of attribution score in attribution patching (Syed et al., 2023), we can define the contribution of an edge $e : n_1 \rightarrow n_2$ with $n_1$ as the upstream node and $n_2$ as the downstream node. We use the indirect effect $IE$ to measure the change in the output caused by the patching of the edge. Thus given a dataset $\mathcal{X}$ for patching, the contribution of edge $e$ can be expressed as follows:

$$
\begin{aligned}
c(e) &= \mathbb{E}_{x_i \sim \mathcal{X}} \big[ |c_i(e)| \big] \\
&= \mathbb{E}_{x_i \sim \mathcal{X}} \big[ |IE(e; x_i)| \big]
\end{aligned}
\tag{4}
$$

Note that the contribution of edge $e$ is directly reflected in the change of the final output (the logit of the language model, etc.), which is the indirect effect caused by the change of the value in the downstream node $n_2$, while the change of the node $n_2$ is directly caused by the change of the upstream node $n_1$. The difference between the direct effect and the indirect effect is that the former keeps all the other nodes that could influence $n_2$ unchanged and only studies the influence from $n_1$ to $n_2$, while the latter allows all the changes in nodes between $n_2$ and the logit. For more refined definitions for these two concepts, please refer to Pearl (2001).

Therefore, to measure the direct effect from $n_1$ to $n_2$, we set the value of $n_1$ to another value $n_1(x')$ while keeping the all other nodes between $n_1$ and $n_2$ unchanged. The indirect effect of $e$ to the final output is

$$
\begin{aligned}
IE(e; x) &= IE(n_1 \rightarrow n_2; x) \\
&= \mathcal{L}_m \big[ M \big( x \,|\, do(n_2 \leftarrow n_2(x')) \big) \big] - \mathcal{L}_m \big[ M(x) \big]
\end{aligned}
\tag{5}
$$

in which the corrupted value $n_2(x')$ of the downstream node is

$$
n_2(x') = n_2(x) - n_2^{n_1}(x) + n_2^{n_1}(x')
\tag{6}
$$

Equation 6 shows the direct effect $n_2^{n_1}(x') - n_2^{n_1}(x)$ from $n_1$ to $n_2$, where

$$
n_2^{n_1}(x') = n_2^{n_1} \big( x \,|\, do(n_1 \leftarrow n_1(x')) \big)
\tag{7}
$$

To simplify the equation, we apply a first-order Taylor expansion to $IE$ at $n_2 = n_2(x)$, then

$$
IE(e; x) \approx \mathcal{L}_m[M(x)] + \big[ n_2(x') - n_2(x) \big]^\top \nabla_{n_2} \mathcal{L}_m[M(x)] \big|_{n_2(x)} - \mathcal{L}_m[M(x)]
\tag{8}
$$

$$
= \big[ n_2(x') - n_2(x) \big]^\top \nabla_{n_2} \mathcal{L}_m[M(x)] \big|_{n_2(x)}
\tag{9}
$$

Thus we have

$$
IE(e; x) = \big[ n_2(x) - n_2^{n_1}(x) + n_2^{n_1}(x') - n_2(x) \big]^\top \nabla_{n_2} \mathcal{L}_m[M(x)] \big|_{n_2(x)}
$$

$$
= \big[ n_2^{n_1}(x') - n_2^{n_1}(x) \big]^\top \nabla_{n_2} \mathcal{L}_m[M(x)] \big|_{n_2(x)}
\tag{10}
$$

To further simplify Equation 10, we apply another Taylor expansion at $n_1 = n_1(x)$ to $n_2^{n_1}$. Then we have

$$
IE(e; x) \approx \Big\{ n_2^{n_1}(x) + \big[ n_1(x') - n_1(x) \big]^\top \nabla_{n_1} n_2^{n_1} \big|_{n_1(x)} - n_2^{n_1}(x) \Big\} \cdot \nabla_{n_2} \mathcal{L}_m[M(x)] \big|_{n_2(x)}
$$

$$
= \big[ n_1(x') - n_1(x) \big]^\top \nabla_{n_1} n_2^{n_1} \big|_{n_1(x)} \nabla_{n_2} \mathcal{L}_m[M(x)] \big|_{n_2(x)}
\tag{11}
$$

Thus we come to the final form of edge contribution in Equation 11. Our derivation takes the simplest situation into consideration, while it works well in practice. For more discussions on the relevant topic, please refer to Kramár et al. (2024). For implementation details, please refer to our code.

# E    DETAILS FOR THE SUBJECT-VERB DISAGREEMENT TASK

To be brief, the goal of this task is to change the grammar in a language model from (a) to (b) as follows:

> (a) *We apologize, but this video has failed to load.*
> (b) *We apologizes, but this video have failed to load.*

In the example above, *We* and *this video* are subjects, *apologize* / *apologizes* and *has* / *have* are verbs, and *We* and *video* are also called the END tokens that appear before the verbs. The change from *apologize* to *apologizes* or from *has* to *have* is called a flip.

## E.1    DATA PREPARATION

We use the first 10k samples from Pile Gao et al. (2020), which consists of 22 smaller, high-quality datasets. Firstly, in order to get relatively simple and clean sentences, we filter out the content in Github, ArXiv, PubMed Abstracts, PubMed Central, StackExchange, USPTO Backgrounds, Pile-CC, DM Mathematics, and FreeLaw. Thus we do not include code or complex formulas in our data. Secondly, we split the corpus with periods '.' as intervals. We remove links to websites, images, and other files. We also remove sentences that are too short (less than 25 characters). Thirdly, we leave only the sentences in the present tense in English and ensure that each sentence is a complete sentence with a punctuation like '.', '?', or '!' at the end. Finally, for each verb in the present tense in a sentence, we convert it to its opposite form. That is, we convert a verb with a part of speech VBP like *go* to VBZ like *goes*, and vice versa. For *be (am / is / are)*, we flip them following: *am → is, is → are, are → am*.

We collect 60,000 samples in total. For experiments, we only use half of the data, which is further split for training (2.4w), validation (3k), and test (3k). All the details for data preparation can be found in our code.

## E.2    EXPERIMENT SETTINGS

We use GPT2-small Radford et al. (2019) for this task. GPT2-small is a decoder-only transformer with 12 layers and 12 attention heads per attention layer. We set the output of the attention and the MLP in each layer as upstream nodes, and the input of the query, key, value, and the MLP in each layer as downstream nodes. This is because the query, key, and value input for an attention head can only affect downstream nodes via the attention output of that head, so the upstream nodes can only be attention head outputs, which is also discussed in Kramár et al. (2024). The parameters to update correspond to the upstream and downstream nodes at both ends of the edges, as discussed in Section 3.2. Details are shown in Table 3.

Table 3: The settings of the nodes and their corresponding parameters in the subject-verb disagreement task. The number of layers $L = 12$ and the number of attention heads in each layer $H = 12$. The notations for parameters are detailed in Appendix C.

| Nodes | Upstream | | Downstream | |
|---|---|---|---|---|
| | $Attn_i^j(x)$ | $MLP_i(x)$ | $x_{Q/K/V}^{i,j}$ | $x_{in}^i$ |
| Parameters | $W_O^{i,j}$ | $W_{out}^i$ | $W_{Q/K/V}^{i,j}$ | $W_{in}^i$ |
| Range | $i \in [0, L), j \in [0, H)$ | | | |

For all experiments, we set the learning rate to 1e-3, and batch size to 16. We use mini-batch SGD with a momentum equal to 0.9 as the optimizer. $K$ is set to 1, which means we perform graph pruning right after an optimization. Each model is trained for 3 epochs, with 100 steps in the beginning for warmup. During training, we evaluate on the valid set every 100 steps. The metric $\mathcal{L}_m$ for measuring the flip from subject-verb agreement to disagreement is the logit difference at the END token, which is:

$$\mathcal{L}_m = logit(W_{v_{flip}}|W_{END}) - logit(W_v|W_{END})$$

where $W$ denotes the tokens in a sentence. For example, the case "We apologize, but this video has failed to load." contains two logit differences: $logit(apologizes|We) - logit(apologize|We)$ and $logit(have|video) - logit(has|video)$. In practice, we only consider the verbs that are tokenized as a single token.

As for the calculation of edge contribution, we follow Syed et al. (2023) and use mean ablation when patching a node. That is to say, for each activation of shape (`batch_size`, `seq_len`, `d_model`), we replace the value at the END token position in each sample with the mean value of all tokens in all samples in a batch.

Experiments are conducted on $4 \times$ A40 Nvidia GPUs.

### E.3  ANALYSIS OF THE QUALITY OF THE DISCOVERED CIRCUIT

As discussed in Section 4.3, the performance cannot be further improved at $N = 1000$, where $N$ is the number of edges saved in circuit discovery. To show this intuitively, the changes with $N$ of the logit difference, PPL, and the ratio of the trainable parameters are illustrated in Figure 7. We can observe that there is a sharp turning point at $N = 1000$, where the curves start to be flat. This serves as a sign that there does exist a circuit that includes all the parameters responsible for the subject-verb disagreement task.

To better prove this conclusion and demonstrate the high quality of the circuit found in our method, we provide another two experiments below.

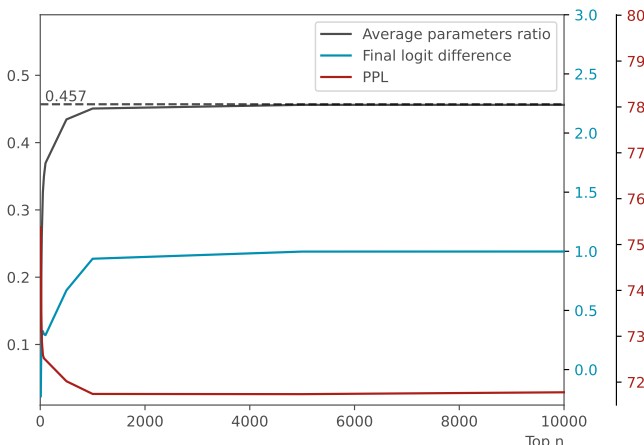

Figure 7: The influence from the setting of top $N$ edges.

### E.3.1  FAITHFULNESS AND COMPLETENESS

Faithfulness and completeness examine a circuit from two different views. Faithfulness tells how much performance a circuit gets, while completeness tells how much performance a circuit fails to capture. Consider a model $M$ with its computational graph $\mathcal{G}$, a circuit $\mathcal{C}$ for a specific task $T$ and a metric $\mathcal{L}_m$ for measuring the output of the model, following the definition in Marks et al. (2024), the faithfulness of the circuit $\mathcal{C}$ is

$$\frac{\mathcal{L}_m[M(\mathcal{C})] - \mathcal{L}_m[M(\varnothing)]}{\mathcal{L}_m[M(\mathcal{G})] - \mathcal{L}_m[M(\varnothing)]} \tag{12}$$

in which $M(*)$ denotes the forward pass of model $M$ with the nodes outside $*$ mean-ablated, and $\varnothing$ denotes an empty circuit. The completeness is defined as

$$\frac{\mathcal{L}_m[M(\mathcal{G}\backslash\mathcal{C})] - \mathcal{L}_m[M(\varnothing)]}{\mathcal{L}_m[M(\mathcal{G})] - \mathcal{L}_m[M(\varnothing)]} \tag{13}$$

where $\mathcal{G}\backslash\mathcal{C}$ denotes the complementary set of circuit $\mathcal{C}$. The completeness of circuit $\mathcal{C}$ is actually the faithfulness of circuit $\mathcal{G}\backslash\mathcal{C}$.

In practice, we calculate the faithfulness and completeness of the circuits for subject-verb disagreement at $N = 100, 500, 1000, 5000, 10000$ edges. Results are shown in Figure 8. It can be seen that $N = 1000$ also serves as a turning point for the curves of faithfulness and completeness. The faithfulness of the circuits remains relatively high after $N = 1000$, ensuring the high quality of circuits.

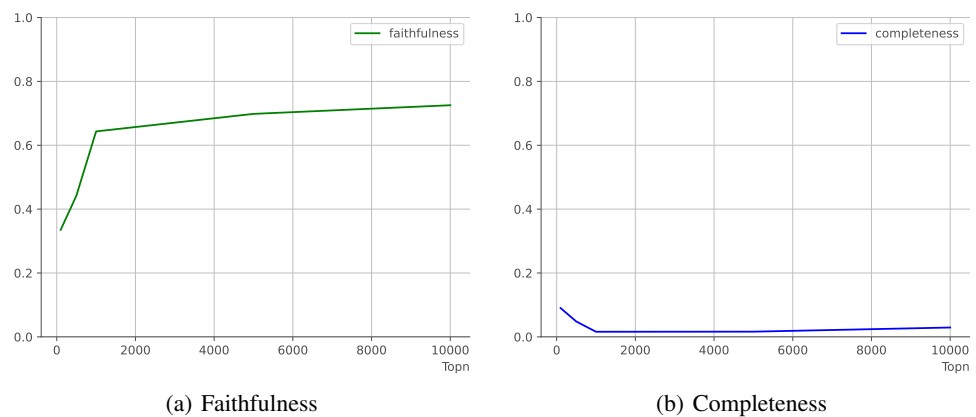

(a) Faithfulness  (b) Completeness

Figure 8: The faithfulness and completeness of the circuits for subject-verb disagreement.

### E.3.2 ABLATION STUDY OF RANDOM ACTIVATION

Random activation means during training, we randomly unfreeze some parameters outside the circuit. Since we assume that the circuit with $N = 1000$ edges already includes all needed parameters for the subject-verb disagreement task, we randomly select a part of the parameters outside the $N = 1000$ circuit and involve them in optimization. In practice, we randomly activate 10%, 20%, 30% and 40% of the outside parameters, and compare the results with before. Results are shown in Figure 9. When 10% of the parameters outside the circuit are activated, the result is almost the same as before. When the ratio gets larger, we observe that the PPL is higher than before when random activation is performed, though the logit difference increases. This is because when extra parameters are summoned to fit the new data distribution, the original functions corresponding to those parameters may be destroyed. Thus the performance is improved at the expense of harming other abilities. Therefore, the circuit we find at $N = 1000$ edges is almost the exact circuit for subject-verb disagreement.

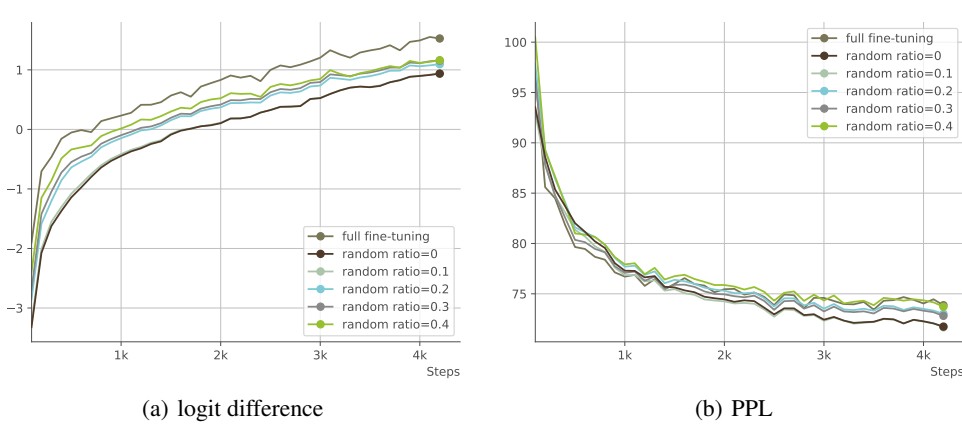

(a) logit difference  (b) PPL

Figure 9: Experiment results of random activation.

### E.4 ANALYSES OF THE TRAINING DYNAMICS

#### E.4.1 INTERPRET THE CIRCUIT FOR THE SUBJECT-VERB DISAGREEMENT TASK

To interpret the circuit for the subject-verb disagreement task, we first analyze this task from the human perspective. To decide the form of a verb, we need to (i) find out the subject in the context and (ii) adjust the form of the verb according to the attributes of the subject, including the person attribute and the number attribute. Therefore, we assume that there exist at least two kinds of attention heads responsible for the two functions above respectively. We name the two kinds of attention heads as the **Subject Identification Heads** and the **Subject Attribute Heads**. Note that we only focus on self-attention instead of MLP because only attention layers move information across tokens, which is important for completing this task. Besides, each of the whole MLP layers is regarded as a node in the circuit in our experiment, thus we do not expect to figure out any specific function from it.

Next, we look for the two kinds of heads discussed above.

**Subject Identification Heads**    To find out the Subject Identification Heads, we check the attention pattern of each attention head at the END token since the END token is directly used for predicting the verb token. We sort the heads in descending order according to their attention weights from the END token (query) to the subject tokens (key). Then we keep the heads in which the attention is mainly paid to the subject in the context.

We find that there exist two types of Subject Identification Heads. The type I heads mainly attend to the last token itself at the END token, so the attention pattern is a diagonal line. This type of head is helpful when the subject is exactly the END token, e.g. "He is ...", "The girls are ...", etc. The type II heads attend to the subject which is several tokens before the END token, e.g. "The kid who is holding an ice cream in hand is ...", "The famous scientist, who is also an artist, has ...", etc. The type II heads obviously have the ability of syntactic analysis and subject identification, while the type I heads may just happen to attend to the subject that overlaps with the last token.

The type I Subject Identification Heads in GPT2-small includes head.0.1, head.0.3, head.0.5, etc., while the type II heads include head.8.5, head.10.5, head.10.9, head.11.8, etc. The attention patterns for both kinds of heads are shown in Figure 10(a) and Figure 10(b) respectively.

It is worth noticing that the Subject Identification Heads remain unchanged over the training process, which means their function is preserved and shared between subject-verb agreement and subject-verb disagreement.

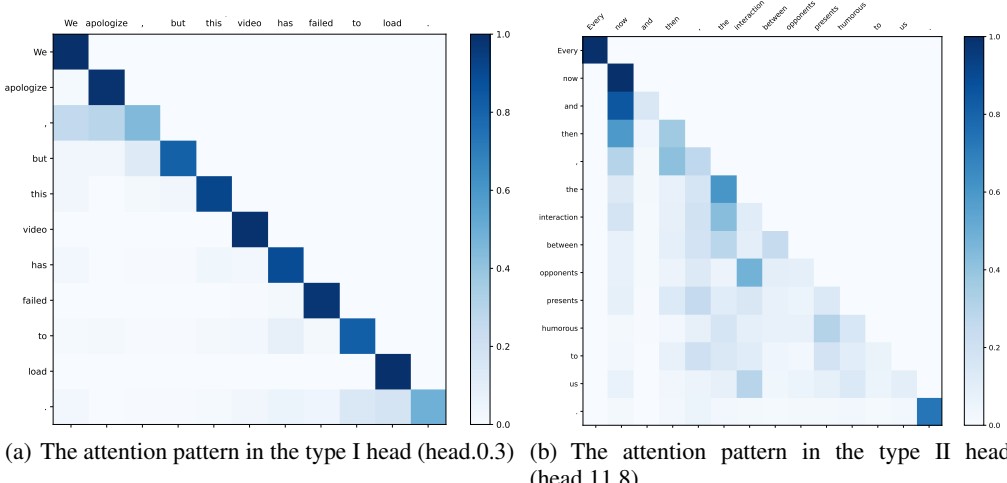

(a) The attention pattern in the type I head (head.0.3)    (b) The attention pattern in the type II head (head.11.8)

Figure 10: The examples of the attention patterns in the Subject Attribute Heads. In Figure 10(a), the END token "*video*" attends to the subject "*video*" which is also the END token itself. In Figure 10(b), the END token "*opponents*" mainly attends to the subject "*interaction*".

**Subject Attribute Heads**    To find out the Subject Attribute Heads, we need to find out which heads directly affect the match between the verb and the subject, which is measured by the logit difference between the flipped verb and the original verb at the END token. Suppose the output of the final layer at the END token is $x_{END} \in \mathbb{R}^D$, then the logit difference is $(W_{U(v_{flip})} - W_{U(v)}) \cdot x_{END}$, in which $W_{U(v_{flip})} - W_{U(v)}$ is called the logit lens nostalgebraist (2020). Since the logit lens is fixed, we expect the projection of $x_{END}$ on the direction of the logit lens to be large, thus encouraging the probability difference between the two opposite verb forms (love v.s. loves, etc.). As discussed in Appendix C, the output $x_{END}$ which is in the residual stream can be decomposed into the linear addition of the outputs from the previous layers. Therefore, the output of a Subject Attribute Head is a part of $x_{END}$ and would encourage the value of the logit difference. Thus, a Subject Attribute Head is an attention head that has a large dot product value with the logit lens.

In practice, we calculate the dot product between the logit lens $W_{U(v_{flip})} - W_{U(v)}$ and the output $Attn_i^j(x)$ from each attention head in each layer over a batch of samples. The result is shown in the main text in Figure 3. The darker the color, the larger the absolute value of the dot product is, which implies that the head is more likely to be a Subject Attribute Head. We observe that head.6.0, head.6.5, head.8.5, head.10.9, and so on see obvious flips (Figure 3(a)) from positive to negative or the opposite direction, which implies that they are directly responsible for the match between the subject and the verb. During training, the parameters inside these heads adjust themselves to the new data distribution, while their function type remains unchanged, which is an interesting finding of the self-regulation ability inside the model.

**Collaborative Heads**    Finally, we notice that some of the Subject Attribute Heads (head.8.5, head.10.9, etc.) are also Subject Identification Heads, which means they also attend to subject tokens. We wonder if there exist some heads in previous layers that could influence the behavior of the Subject Attribute Heads. That is to say, the Subject Attribute Heads do not act alone but collaborate with other heads.

To find out these heads, we knock out the upstream heads one at a time at the END token using mean ablation. We observe the change in the attention pattern of each Subject Attribute Head and then keep the heads that bring obvious changes in the attention patterns. In practice, we focus on subject-verb agreement and only check the influence on head.8.5 and head.10.9. We also provide two types of data, corresponding to the two cases discussed in Appendix E.4.1, in order to provide a more detailed analysis. Results show that

- When patching on the type I data in which the END token is exactly the subject, head.1.2 and head.2.11 affect both head.8.5 and head.10.9

- When patching on the type II data in which the subject is several tokens before the END token, head.7.4 and head.2.11 affect head.8.5, while head.2.11 affects head.10.9.

- When we mix the two types of data, we find that head.1.3, 0.8, 2.10, and 6.5 affect head.8.5, while head.1.3, 1.4, 1.6, 6.5, 0.8, and 0.9 affect head.10.9. These heads may be responsible for both types of data while not specifically responsible for a certain type of data, so they appear when patching on the mixed data.

We notice that though the influence of the above heads is relatively large, the absolute influence is sometimes quite small. Therefore, we further conduct an experiment in which we patch multiple heads at a time and check the influence of them on head.8.5 and head.10.9. Results show that when upstream heads are patched together, their combined effect is much higher than the individual effect. Thus we call these heads the Collaborative Heads.

### E.4.2    THE INTERACTION AND COLLABORATION INSIDE THE MODEL

From the discussions above, we can see that the nodes inside a circuit complete a task through a cooperative division of labor. We summarize the interaction and collaboration inside the model as follows:

1. Each node is responsible for a single or multiple functions. As discussed in Appendix E.4.1, we have found attention heads that are responsible for identifying the subject in a sentence or adjust the form of a verb according to the attributes of the subject, or both. Each head has its division of labor when completing a task.

2. Some heads directly affect the output, while others cooperate with them to affect the output indirectly. For example, head.8.5 directly matches the verb with the subject, while head.7.4, head.1.3 and so on indirectly affect the output through cooperation with head.8.5.

3. Several nodes achieve a common goal through cooperation. For example, head.1.3, 1.4, 1.6, 6.5, 0.8, and 0.9 affect head.10.9 through combined effect, which means their influence on head.10.9 only appears when they act together.

### E.4.3 THE EVIDENCE OF HEBBIAN LEARNING

As discussed in Section 4.4, we observe that some edges in the circuit are strengthened or weakened during training, just like the Hebbian learning proposed in Do (1949) in neuroscience. As stated by Hebb, a synapse between two neurons is strengthened when the neurons on either side of the synapse have highly correlated outputs, which means they are often activated synchronously. The theory is often concluded as "Cells that fire together, wire together" Shatz (1992). For two neurons $i$ and $j$, a common description of hebbian learning is as follows:

$$w_{ij} = \frac{1}{p} \sum_{k=1}^{p} x_i^k x_j^k \tag{14}$$

where $w_{ij}$ is the weight of the connection between the two neurons, and $x_i^k$ and $x_j^k$ are the $k$-th inputs for $i$ and $j$ respectively. When it comes to the computational graph of a model, the nodes and edges in the graph could be viewed as the neurons and their connections in a brain from the perspective of neuroscience.

During training, we find that some of the edges are obviously stronger than others, which means they have higher edge contributions. Besides, they are strengthened all the way during training. Specifically, we analyze the circuits during training in the subject-verb disagreement task, with the setting of top $N = 1000$ edges. We check the results at 2000, 3000, and 4000 steps respectively, and visualize the circuits with the top 35 edges in Figure 4. Note that the thickness of an edge corresponds to the logarithm of the edge contribution, that is $log[1 + c(e)]$. The details are shown in Table 1.

We also find that the edge contribution may keep decreasing during training. This is quite similar to the self-organization of cells inside human brains, a well-known phenomenon of which is the lateral inhibition among neurons Jacobson (1993), which means an activated neuron can reduce the activity of its neighbors. Inspired by this, Kohonen (2012) developed the self-organizing maps (SOM), an unsupervised algorithm that leverages the Winner-Take-All strategy to perform competitive learning. The core ideas behind SOM are:

- The neurons inside a neural network learn to represent data through competition, i.e. given an input, some neurons are activated while others are inhibited.

- Different inputs are represented in a topologically ordered manner, i.e. different neurons are responsible for different features in a well-organized style.

In our study, we find that the dynamic change of edges echoes the above discussions on competition and self-organization. During training, some connections between nodes are strengthened, which may reduce the intensity of other connections. After training, the components inside a model have reorganized themselves to adapt to the new data distribution. When faced with an input, different regions inside the computational graph are responsible for different subtasks and collaborate to complete a goal, as discussed in Appendix E.4.1 and Appendix E.4.2.

### E.4.4 THE CIRCUITS BEFORE AND AFTER FINE-TUNING

We present the circuits of the subject-verb disagreement task before and after fine-tuning in Figure 11(a) and Figure 11(b) respectively. The circuit of subject-verb disagreement is trained under the setting of $N = 1000$ edges, and for both of the circuits we only present the top 100 edges. It can be seen that most of the heads we discussed in Appendix E.4.1 are included in the circuit. Besides, it is obvious that the circuits before and after fine-tuning share a lot of nodes, which implies that the function shift from subject-verb agreement to disagreement happens mostly in the **polarity change** of nodes, instead of randomly assigning the ability to other nodes. This is an intuitive finding, which

not only demonstrates the rationality of circuit-tuning as well as our analyses but also provides new insights for our understanding of the mechanism inside language models.

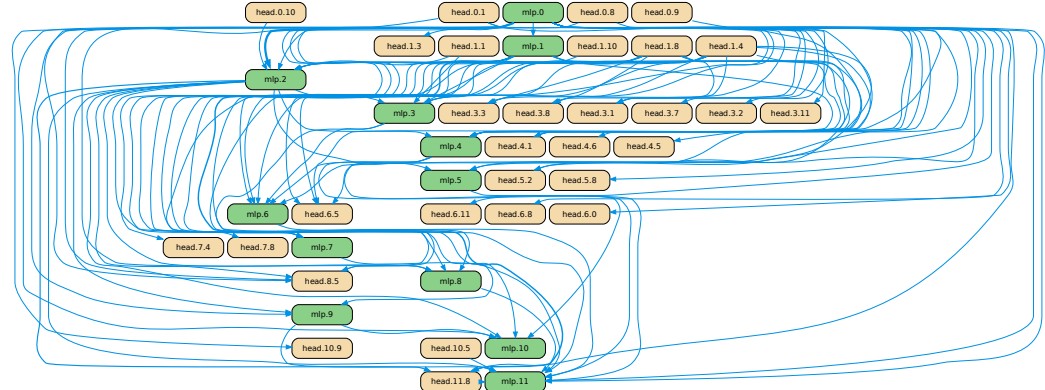

(a) The circuit before fine-tuning (subject-verb agreement).

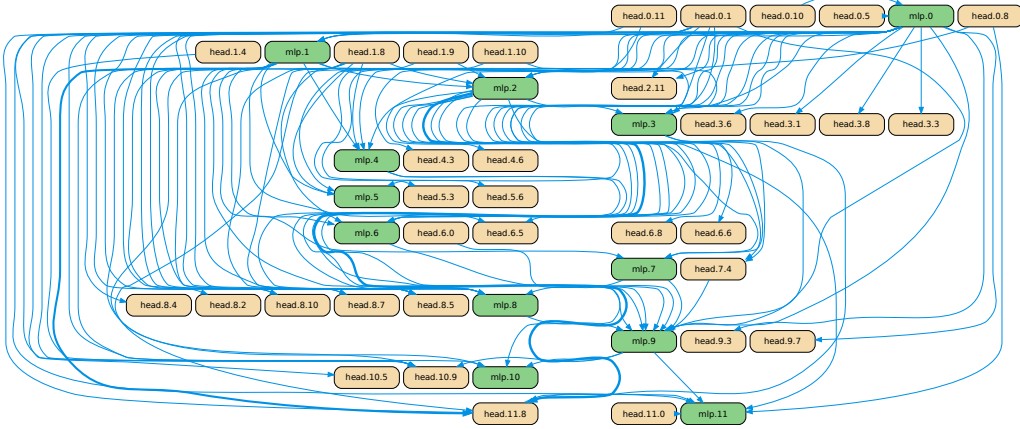

(b) The circuit after fine-tuning (subject-verb disagreement).

Figure 11: The circuits of subject-verb agreement (top) and subject-verb disagreement (bottom).

### E.4.5 THE EXPERIMENT RESULTS AFTER THE REVISION OF ATTRIBUTION SCORE

During the analyses in Appendix E.4.1, we find that the original definition of the attribution score in EAP Syed et al. (2023) fails to capture all the relevant edges in a task. For example, head.6.0 which is a Subject Attribute Head fails to appear in the circuit. We assume that there exists a situation where an important node is connected with many other nodes, but each edge is not that strong. For example, as illustrated in Figure 12, an upstream node $n_a^u$ is connected with four downstream nodes, while another upstream node $n_b^u$ is connected with only one downstream node. Since the edge between $n_b^u$ and $n_5^d$ is stronger than any edge between $n_a^u$ and the nodes connected with it, the edge $n_b^u \rightarrow n_5^d$ may be kept in the circuit, while the edges in $\mathcal{E}_a = \{n_a^u \rightarrow n_i^d | i = 1, 2, 3, 4\}$ may be left out. As a result, $n_a^u$ is not involved in optimization, though the sum of the edge contributions of all edges in $\mathcal{E}_a$ may be almost the same or even larger than that of $n_b^u \rightarrow n_5^d$.

Thus we calculate the edge contribution of an edge $e : n_i^u \rightarrow n_j^d$ as below:

$$c(e)' = c(e) \cdot \sum_{k=1}^{N_{down}^i} c(n_i^u \rightarrow n_k^d) \cdot \sum_{k=1}^{N_{up}^j} c(n_k^u \rightarrow n_j^d) \tag{15}$$

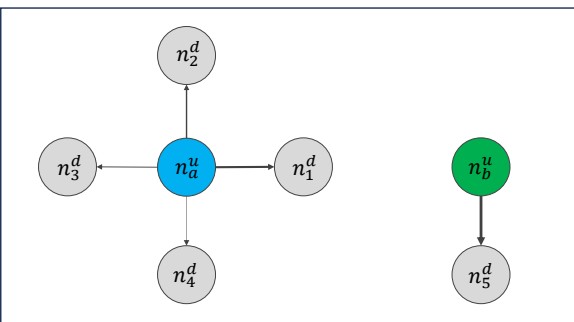

Figure 12: A sketch for the idea behind the revision on attribution score.

where $c(e)$ is the original attribution score in EAP, $N_{down}^i$ is the number of the downstream nodes of $n_i^u$, $N_{up}^j$ is the number of the upstream nodes of $n_i^u$. The revision considers the contributions from all the edges connected to the upstream node and the downstream node. To verify it, we conduct a new experiment on the subject-verb disagreement task and compare it with the result before. Results are shown in 13. Details can be found in Table 4. Compared with the original attribution score, our method improves the logit difference steadily, while even bringing down the computation to some extent.

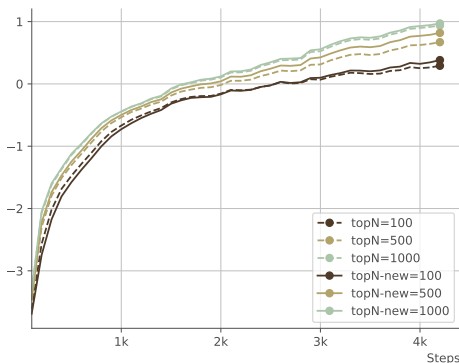

Figure 13: The experiment result after the revision on attribution score.

Table 4: Comparison between the performance before and after the improvement on attribution score.

| Top n & Methods | Original EAP | | | Improved EAP | | |
|---|---|---|---|---|---|---|
| | logit difference | PPL | Avg. Param ratio (%) | logit difference | PPL | Avg. Param ratio (%) |
| 50 | 0.292 | 72.59 | 32.61 | 0.350 | 72.71 | 26.88 |
| 100 | 0.291 | 72.50 | 36.96 | 0.382 | 72.53 | 34.12 |
| 500 | 0.670 | 72.02 | 43.50 | 0.819 | 72.05 | 42.50 |
| 1000 | 0.937 | 71.74 | 45.61 | 0.970 | 71.73 | 45.55 |

# F  DETAILS FOR THE COMPLEX TASKS

## F.1  DETAILS FOR TASK SETTINGS

### F.1.1  REASONING-BASED TASKS

**Mathematics** We use GSM8K Cobbe et al. (2021) as the dataset, which contains about 8.5k grade school math problems with natural language solutions. The answer to a problem not only contains the final answer but also provides the process for solving the problem. An example of this task is shown in Table 5.

Table 5: An example in the GSM8K dataset.

| Question | Answer |
|---|---|
| Janet's ducks lay 16 eggs per day. She eats three for breakfast every morning and bakes muffins for her friends every day with four. She sells the remainder at the farmers' market daily for $2 per fresh duck egg. How much in dollars does she make every day at the farmers' market? | Janet sells $16 - 3 - 4 = << 16 - 3 - 4 = 9 >>$ 9 duck eggs a day. She makes $9 * 2 = \$ << 9 * 2 = 18 >>$ 18 every day at the farmer's market. ####18 |

During training, the NLL loss also serves as the metric $\mathcal{L}_m$ for measuring the output of the model. For evaluation, we use Acc@1 as the metric, which means for each problem in the test set we only sample one answer from the model.

**Logical Reasoning** We use Contexthub Hua et al. (2024) as the dataset, which consists of problems of 4 difficulty levels, including deductive and abductive reasoning in 12 distinct categories or domains from Wikipedia. The problems together with their reasoning processes are instantiated automatically by LLMs following the fixed formal logic templates. The whole dataset contains 18,240 samples. We only use level 1 and level 2 in our experiment for convenience, which include 6720 samples in total. An example of this task is shown in Table 6.

Table 6: An example in the Contexthub dataset.

| Item | Template | Instantiation |
|---|---|---|
| Premise | \<aaa\> | The Sahara desert receives heavy rainfall this year. |
| | \<aab\> | The Amazon rainforest experiences severe drought conditions. |
| | \<aac\> | Some of Earth's major ecosystems are undergoing significant changes in weather patterns. |
| Question | (aaa OR aab) → aac. Given aac is False, what is the value of aab? | If either the Sahara desert receives heavy rainfall this year or the Amazon rainforest experiences severe drought conditions, then it implies that some of Earth's major ecosystems are undergoing significant changes in weather patterns. Given that it is false that some of Earth's major ecosystems are undergoing significant changes in weather patterns, what can be determined about the Amazon rainforest experiencing severe drought conditions this year? (True, False, or N/A (undetermined). |
| Reasoning | (aaa OR aab) → aac = False. Given aac is False, the value of premise (aaa OR aab) is False, thus, the value of aab is abduced as False. Thus, the answer is False | "The Sahara desert receives heavy rainfall this year" or "The Amazon rainforest experiences severe drought conditions") logically implies "Some of Earth's major ecosystems are undergoing significant changes in weather patterns" whose corresponding truth value is False. Given "Some of Earth's major ecosystems are undergoing significant changes in weather patterns" is False, the value of premise ("The Sahara desert receives heavy rainfall this year" or "The Amazon rainforest experiences severe drought conditions") is False, thus, the value of "The Amazon rainforest experiences severe drought conditionsïs abduced as False. Thus, the answer is \<answer\>False\</answer\> |

During training, the NLL loss also serves as the metric $\mathcal{L}_m$ for measuring the output of the model. For evaluation, we use the average F1 score over all categories of data.

For all reasoning-based tasks, we add an instruction "Please answer step by step." at the end of the question in order to guide the model to answer the question step by step.

### F.1.2 REASONING-FREE TASKS

**Gender De-biasing** According to Gallegos et al. (2024), there are various kinds of expressions in social bias. In this study, we focus on the gender bias in occupations. We aims to break down the binary gender stereotype of a model. For example, given a sentence "the doctor put on [PRP] coat" where [PRP] is a possessive pronoun, we expect the model to choose *his* or *her* with equal probabilities.

During fine-tuning, the model learns to predict the next word in an auto-regressive way, thus we expect the model to balance the probabilities between male attribute words (he/his/him/himself) and female attribute words (she/her/herself) at the END token when generating the next token. Therefore, we use the logit difference between the male attribute words and female attribute words at the END token as the metric $\mathcal{L}_m$ for measuring the output of the model. Specifically, for each sample, we calculate the logit difference between the pronoun and the anti-pronoun, which is the pronoun in the opposite gender. For example, in the case "the doctor put on [PRP] coat", the logit difference is $logit(W_{her}|W_{END}) - logit(W_{his}|W_{END})$, where $W_{END}$ is the END token *on*.

We use BUG Levy et al. (2021) for training, which is a large-scale dataset of sentences sampled from real-world corpora. Each sentence is marked with an occupation and the pronouns referring to it. In practice, we use the "balanced BUG" provided in the dataset which includes 2.5w sentences randomly sampled from BUG to ensure balance between male and female entities and between stereotypical and non-stereotypical gender role assignments. We perform coreference resolution ourselves to filter out the samples in which the coreference is not right, and leave 1.5w samples for training, which contains 151 types of occupations and each sentence only contains one (occupation, pronoun) pair.

Though the training set we use is balanced between genders and stereotypes, the number of samples for each occupation is not balanced. To further improve performance, we additionally add a regularization term to the original NLL loss. Then the total loss is

$$\mathcal{L} = \mathcal{L}_{NLL} + \beta \cdot |logit(W_{pron}|W_{END}) - logit(W_{anti-pron}|W_{END})| \tag{16}$$

in which $\beta$ is a hyper-parameter for controlling the weight of regularization. The absolute value of the logit difference aims to minimize the difference between genders in the model's stereotype.

For evaluation, we use WinoBias Zhao et al. (2018) which is a classic dataset for coreference resolution focused on gender bias. There are two types of sentences in WinoBias. The Type 1 sentences require world knowledge related to the context to perform coreference resolution, e.g. "The farmer knows [the editor] because [he] is really famous". The Type 2 sentences can be resolved using syntactic information, e.g. "The CEO called [the hairdresser] and paid [her] over the phone". In practice, we only use the Type 2 sentences to avoid ambiguity. The test set contains 40 types of occupations in total.

To better evaluate the performance of gender de-biasing, we adopt the concept of prejudice risk from Liu et al. (2024) which is used to measure the stereotype in large language models. Specifically, given an occupation $x \in X$, a binary gender attribute $y \in \{male, female\}$ and a context $c \in C$, the stereotype of a model $M$ against $x$ about $y$ in the context $c$ is

$$s_{y|x}^M(c) = \frac{p_{y|x}^M(c)}{p_{y|x}^*(c)} - 1 \tag{17}$$

where $p_{y|x}^*(c)$ is the attribute prediction probability of the unbiased model, thus $p_{y|x}^*(c) = 0.5$ when binary gender is considered. The definition of prejudice risk is

$$R^p = \mathbb{E}_{x \sim X}(r_x^p) \tag{18}$$

where $r_x^p = J(\mathbb{E}_{c \sim C}(s_{y|x}^M(c))$ is the prejudice risk of one occupation $x$, and $J(s_{y|x}^M(c)) = \max_{y \in Y}\{max\{s_{y|x}^M(c), 0\}\}$ is the discrimination risk criterion. For details, please refer to Liu et al. (2024).

**Reading Comprehension** We use SQuAD 2.0 Rajpurkar et al. (2018) as the dataset. The input contains a paragraph from a passage, and a question related to that paragraph. The answer could be a word or a phrase in the paragraph, or "<No Answer >" which means the answer cannot be found in the paragraph. An example for this task is shown in Table 7.

Table 7: An example in the SQuAD 2.0 dataset.

| Item | Content |
|---|---|
| Paragraph | The Normans (Norman: Nourmands; French: Normands; Latin: Normanni) were the people who in the 10th and 11th centuries gave their name to Normandy, a region in France. They were descended from Norse ("Norman" comes from "Norseman") raiders and pirates from Denmark, Iceland and Norway who, under their leader Rollo, agreed to swear fealty to King Charles III of West Francia. Through generations of assimilation and mixing with the native Frankish and Roman-Gaulish populations, their descendants would gradually merge with the Carolingian-based cultures of West Francia. The distinct cultural and ethnic identity of the Normans emerged initially in the first half of the 10th century, and it continued to evolve over the succeeding centuries. |
| Question | In what country is Normandy located? |
| Answer | France |

During training, the NLL loss serves as the metric $\mathcal{L}_m$ for measuring the output of the model, since the answer is a segment of text which may contain one or multiple tokens. For evaluation, we use the development set for convenience, and the metric is exact match and F1 score.

### F.2 DETAILS FOR IMPLEMENTATION

We use Llama-3.2-1B-Instruct, Llama-3.2-3B-Instruct and Llama-3.1-8B-Instruct Dubey et al. (2024) for this task. We set the output of the attention and the MLP in each layer as upstream nodes, and the input of the query, key, value, and the MLP in each layer as downstream nodes. Different from GPT2-small, an MLP layer in Llama is too big to be a node, so we split the input and output of each MLP layer into 64-dimensional MLP heads. Details are shown in Table 8.

Table 8: The settings of the nodes and their corresponding parameters in the complex tasks. For model sizes 1B/3B/8B, the number of layers $L = 16/28/32$, and the number of attention heads in each layer $H = 32/24/32$, and the number of MLP heads in each layer $H^* = 128/128/224$. The notations for parameters are detailed in Appendix C.

| Nodes | Upstream | | Downstream | | |
|---|---|---|---|---|---|
| | $Attn_i^j(x)$ | $MLP_i^k(x)$ | $x_{Q/K/V}^{i,j}$ | $x_{pre}^{i,k}$ | $x_{in}^{i,k}$ |
| Parameters | $W_O^{i,j}$ | $W_{out}^{i,k}$ | $W_{Q/K/V}^{i,j}$ | $W_{gate}^{i,k}$ | $W_{in}^{i,k}$ |
| Range | $i \in [0, L), j \in [0, H), k \in [0, H^*)$ | | | | |

As for the calculation of edge contribution, we use mean ablation as before when patching a node. For reasoning-based tasks and the reading comprehension task in reasoning-free tasks, there is no such thing as the END token. Therefore, for each activation of shape (`batch_size, seq_len, n_head, d_model`), we take all tokens into consideration and use the mean value over all tokens and all samples for mean ablation. For implementation details, please refer to our source code.

The batch size is set to 16 in all experiments. We set the learning rate to 3e-5 for the mathematics and reading comprehension tasks, and 1e-4 for other tasks. Mini-batch SGD with a momentum equal to 0.9 is used as the optimizer. During training, we perform circuit discovery every 8 steps after optimization for efficiency, which is different from the experiments on GPT2-small in which we perform circuit discovery rightly after an iteration step. For each task, we train the model until performance cannot be further improved.

For LoRA, we set $r = 32, \alpha = 64$ for all experiments, since this is the best setting we could get. We have swept a wide range of values for rank $r$ and alpha $\alpha$, and find that the performance cannot be further improved or even decreases when $r$ increased over 32. Actually, Hu et al. (2021) found similar phenomenon when they studied LoRA. From our opinion, this is because LoRA falis to accurately figure out the key parameters needed to be fine-tuned, since all the parameters are changed after LoRA fine-tuning.

For full fine-tuning and LoRA, we use the same optimizer as that in circuit-tuning. For all other hyper-parameters such as learning rate, batch size, training steps, and so on, we just sweep over a range of choices and choose the best ones.

In practice, we find that circuit-tuning is much more stable than full fine-tuning and LoRA. When we sweep over a range of hyper-parameters, we notice that full fine-tuning and LoRA are quite sensitive to the change of learning rate, batch size, training steps, and so on. When it comes to circuit-tuning, the change in evaluation result is relatively moderate while still maintaining good performance.

Experiments are conducted on $8 \times$ A800 Nvidia GPUs.

### F.3 DETAILS FOR EVALUATIONS ON GENERAL CAPABILITIES

To demonstrate that our method is good at preserving general capabilities, we test the fine-tuned models on a set of benchmarks involving general capabilities as well as other capabilities.

For general capabilities, we use MMLU Hendrycks et al. (2020), Winogrande Sakaguchi et al. (2021) and IFEval Zhou et al. (2023). For MMLU, the evaluation metric is the average accuracy over all categories. For Winogrande, we use the development set for convenience, and the evaluation metric is accuracy. For IFEval which is to test the instruction following ability of a model, each prompt contains one or multiple verifiable instructions, thus the evaluation metric is divided into the prompt-level accuracy and instruction-level accuracy. Due to the randomness of generation, each response is tested under multiple transformations, thus the metric is further divided into strict criterion and loose criterion. In practice, we use the prompt-level and instruction-level accuracy averaged on the strict and loose criteria.

For other capabilities, we consider reasoning, coding, and multilingual capabilities. For reasoning, we use GPQA Rein et al. (2023) with accuracy as the metric. For coding, we use HumanEval Chen et al. (2021) with pass@1 as the metric. For multilingual capability, we use MGSM Shi et al. (2022) with the accuracy averaged on all languages.

To check if the general capabilities as well as other capabilities are affected after fine-tuning, we compute the relative change in performance for each capability. For example, if the evaluation results before and after fine-tuning are $x$ and $x'$, then the relative change is $\frac{x'-x}{x}$. In practice, we test each model on each benchmark for 10 times.

