# OpenReview forum: "Fine-Tuning is Subgraph Search: A New Lens on Learning Dynamics"
_ICLR.cc/2026/Conference — ICLR 2026 Conference Withdrawn Submission_

### Official Review · Reviewer_MVC6 · 2025-10-26

**Soundness:** 1
**Presentation:** 1
**Contribution:** 2
**Rating:** 2
**Confidence:** 3

**Summary:**

The submission a method for restricting gradient updates to a relevant circuit during fine-tuning.

I did not read the paper very carefully, because I found the poor quality of the writing made it difficult.

From Algorithm 1 (page 4), it seems the method is simply:
1) run SGD, but only update the parameters of a particular circuit C
2) every K gradient updates, run an pre-existing circuit discovery method to discover a new circuit C
although this seems somewhat contrary to the description in the abstract of it iteratively building a subgraph (which).

Section 4 experiments focus on interpreting models before and after fine-tuning (but not, that I saw, *during* training).

Section 5 presents another set of experiments where the method outperforms LORA.

**Strengths:**

The proposed method is simple and elegant.

The finding that restricting fine-tuning to a relevant circuit can provide utility is interesting, and may be novel (I've spoken to several researchers who intended to work on something like this about 2 years ago, but I didn’t find any works that obviously actually did this in their Google scholar, so I’m not sure).  Indeed, the results in Table 1 appear quite strong, and this is noteworthy and potentially enough to merit publication absent other issues with the work.  However, at present, the submission does not do enough work to convince me of the significance of this result; I would expect a thorough review of related work that seeks to accomplish effective and efficient fine-tuning, and comparison to more methods, not just LoRA.  I'd also want to be convinced that the fine-tuning hyperparameters are well-tuned.

**Weaknesses:**

The clarity and quality of presentation is low overall, and includes various imprecise claims, some of which seem false.

From what I can tell, the analysis in Section 4 fails to make any use of the Cs that are discovered throughout training, and to only perform static analyses on the model before and after fine-tuning.  So while the method could be used to study the dynamics of learning, as promised elsewhere, it seems the submission never actually does that.  This invalidates much of the motivation of the work and claims to novelty.

Overall, I didn't find the analyis in Section 4 very cogent overall.  The findings mostly seem pretty banal, and I'm not sure this section has much to add over previous works such as the references Wang et al. (2022).

The abstract refers to a hypothesis but does not state one clearly or precisely.  This sort of imprecise language is an issue throughout the submission, e.g. I don’t really know how to evaluate claims like “there lacks a general and unified way or pipeline for interpreting the
learning process that is applicable to various scenarios from the view of mechanistic interpretability” or “We believe that the learning and generalization process of the model on a specific task is to find and optimize the subgraph in a computational graph.”  The hypothesis described on 148-149 is not precise enough, either.  What does the claim that a model is “dynamically search[ing] a subgraph” mean?

The introduction is quite weak overall.  Besides the two paragraphs on related work at the outset, I’m not able to extract much information from it beyond what was in the abstract; the content seems highly redundant.

The submission spends a lot of time talking about how to view a model as a computational graph and describing subgraphs; this is unnecessary, as all of this is well-established in previous work.  I’d prefer if the work just called the subgraphs “circuits” consistently throughout.

**Questions:**

Do you perform any analysis of how C changes throughout learning?

Did you experiment much with different values of K?  Is it important that C is changed throughout the fine-tuning process (vs. fixed at the outset)?

---

> ### Author Response · Authors · 2025-11-19
>
> Thanks a lot for your valuable suggestions, especially those concerning the writing~
> We'll try our best to answer your concerns!
>
> **1. Analysis of how circuits changes throughout learning**
>
> Thanks for pointing this out. Here we would like to answer your concerns from the following two aspects:
> * *The dynamic analysis you may want to see*:
>   * Perhaps what you are interested in is in *Finding 5* of Section 4.4. As shown in Figure 4, we captured circuits at multiple training steps during fine-tuning and observed the changes in the strength (edge contribution) of the edges, discovering a phenomenon similar to Hebbian Learning. With this observation, we provide a deeper discussion on the mechanism of circuit-tuning and compared the differences between static and dynamic analysis methods. We hope this discussion is what you were looking for.
>   * We sincerely apologize that, due to page limitations, a lot of analysis regarding this part was moved to the Appendix, which brings confusion and difficulty to your reading. We guarantee that we'll increase the discussion of learning  dynamics in the main body of our paper~ Thanks again for your advice.
> * *Dynamic analysis in Finding 2 & 3*:
>   * The comparison between the circuits before and after fine-tuning gives us an overview of the how a model's internal changes during fine-tuning. Even if it is only focused on the two endpoints (beginning and end of training), we can still explore the alterations within the model's internal components through a before-and-after comparison, which is interesting and meaningful in our view.
>
> **2. The writing of this paper**
>
> First, we sincerely apologize that some expressions in the paper caused confusion in your reading, especially in the introduction, where our overly simplistic narrative failed to convey our ideas precisely. To address this, we will reorganize the background of this work and include more rigorous language to improve the reading experience!
>
> As for the "hypothesis", we have reflected on our work and decided to refer to it as a new fine-tuning paradigm. We'll make it more rigorous.. During our writing, we intended to first put forward an intuitive idea, and then lead the reader step-by-step through the inner meaning of circuit-tuning in subsequent sections. While we confess it's not rigorous, so sorry for that!
>
> In previous versions of the paper, we attempted to formalize this in a theoretical style, but reviewers generally considered that unnecessary. Therefore, for this submission, we adopted a more relaxed narrative style, hoping to help readers understand the concept by starting simple and gradually moving to depth.
> As feedback from different reviewers varied greatly, it is challenging to reach a version that everyone could read smoothly. We hope you can understand this difficulty. Nonetheless, we are very glad you offered new suggestions and we'll revise our paper!
>
> **3. The contribution of our work**
>
> Our main contribution is showing that one can study circuits from a dynamic view. The idea has similar practices in previous work on continual learning, but is rarely attempted in the field of mechanistic interpretability. We hope this work can provide inspiration to the community.
>
> Regarding your concerns, we would like to discuss two points here:
> * *Comparison with the work of Wang et al (2022)*.: They conducted a thorough and brilliant analysis of the IOI task, providing us with a practical analysis paradigm. While our work aims to actively integrate circuit analysis with fine-tuning to observe the dynamic evolution of the circuit from the view of fine-tuning. What excites us the most is our analysis of the learning dynamics during circuit-tuning (we have added more content to the paper regarding this). We believe that this new analysis paradigm and fine-tuning method is promising for the analysis of learning and generalization.
> * *Discussions on other PEFT methods*: We prefer to treat mitigating catastrophic forgetting as an additional property of circuit-tuning, since we are not aimed at efficient fine-tuning. Thus we did not place emphasis on comparing it with other PEFT methods, but we'll add discussions about this part~
>
> **4. About $K$**
>
> Yes, we have tried different values for $K$. In Section 4, since we were experimenting with a small model (GPT-2), we set $K$ to $1$, meaning that a new circuit was found immediately after each optimization step. For larger models at the 7B scale, we set $K$ to a larger value to balance training efficiency and performance. Empirical results show that the larger the value of $K$, the faster the training speed (because the number of graph pruning operations is reduced), but when $K$ is greater than $8$, the convergence speed slows down.
>
> ---
> Sorry again for the lack of clarity. We are happy to receive your advice and revise our paper.
>
> Since a lot of time has been spent on this work, we'll appreciate it if you could re-evaluate it~ Looking forward to your reply !!

---

> ### Comment · Reviewer_MVC6 · 2025-11-19
> **A quick response**
>
> 1) Sorry for missing Finding 5 of Section 4.4.  I agree this is an example of a dynamic circuit analysis effort during circuit-tuning.  However, the result is very unclear to me, since:
> a) The figure is inadequate: I can only distinguish "thick" from "thin" lines, and I don't see anything reflecting the activation of the units, which would be necessary to establish something like Hebbian learning ("fire together --> wire together").
> b) The writing is still very unclear.
>
> 2)  I appreciate that you will make efforts to clarify the writing, and what you've written seems sensible, but I'm not confident it will be enough given how much work is needed.
>
> 3) Unless there is more than finding 5 related to the main intended contribution ("showing that one can study circuits from a dynamic view"), I think it's not nearly enough.  So then I think comparison with Wang et al. or PEFT methods are more apt.  Of these, the current results look stronger to me on the PEFT comparison (and this would fit with the reframing of this as more of a methods paper), but a much more thorough comparison would be needed.
>
> Overall, your response updates me positively a little bit, but for these reasons, I think it would take a lot more for me to support the paper, and I'm not sure I will be able to, unless I learn I've missed or misunderstood more.

---

> > ### Author Response · Authors · 2025-11-24
> >
> > **About Hebbian Learning**
> >
> > Thank you for pointing this out. As you mentioned, Hebbian Learning emphasizes the process of "fire together $\rightarrow$ wire together." We would be happy to further discuss the “fire of nodes". The edge contribution we employed (derived from Edge Attribution Patching, Eq. 11 in Appendix D) essentially calculates the product of the activation value of the upstream node and the gradient flowing from the downstream node.
> >
> > $ c(e) \approx  \mathbb{E}[ [n_{1}(x^{noise})-n_{1}(x^{clean})]^T \nabla_{n_{2}}L[M(x^{clean})] |_{n=n_2(x^{clean})} ] $
> >
> > where $n_{1}$ is an upstream node and $n_{2}$ is a downstream node, $x$ is the clean input and $x’$ is the corrupted input, $M$ is the model and $L$ is the metric to measure the output difference before and after edge patching.
> >
> >  A thick edge in our figure mathematically necessitates that the node is both highly activated and highly relevant to the task loss. If a node fails to “fire”, then the activation value (in case of an upstream node) or the gradient (in case of a downstream node) of it would be relatively small compared with the case where both the upstream and downstream nodes have larger activation values, and then the edge attribution would approach zero. Thus, when an edge is thick (with a relatively large edge contribution), the relevant nodes would also have relatively large activation values.
> >
> > We acknowledge that claiming this is Hebbian learning was too strong given the visual evidence. It is true that Figure 4 only shows the change in edge thickness and does not display the change in node activation values. We have revised the statement in Finding 5 to weaken the comparison with Hebbian Learning. We clarify that the "strengthening" of edges represents the consolidation of task-relevant pathways—where high activation aligns with high task relevance—functionally mirroring the outcome of Hebbian learning. We only use Hebbian Learning as a reference to ensure the rigor of the discussion.

---

> > > ### Author Response · Authors · 2025-11-24
> > > **Continue**
> > >
> > > **For the comparison with other methods**
> > >
> > > We agree with your suggestions. We believe the contributions of this work can be summarized as:
> > >
> > > a) Providing a new research paradigm for the interpretability research community, which allows the study of circuit changes during fine-tuning from a dynamic perspective, thereby better understanding the learning and generalization processes of neural networks.
> > >
> > > b) Circuit-tuning simultaneously comes with the ability to precisely fine-tune, adaptively selecting task-relevant parameters without affecting parameters outside the circuit (i.e., task-irrelevant parts), thus preventing catastrophic forgetting.
> > >
> > > We are happy to provide a deeper discussion on this. We agree with your suggestion to compare our method with **other PEFT methods**. Furthermore, we believe it is also necessary to compare with methods related to **model pruning** to provide you with a more comprehensive analysis.
> > >
> > > **Comparison with PEFT**
> > >
> > > Let's start our discussion with **LoRA**, which is the most fundamental and widely used method.
> > > LoRA performs a global low-rank update ($\Delta W = BA$). Although LoRA limits the parameter update to a low-dimensional space via the two matrices $A$ and $B$, during inference, the parameter increment matrix $\Delta W$ is restored to the original dimension of the model after the multiplication of matrices $A$ and $B$ within the LoRA adapter. This leads to LoRA affecting all parameters in a model, inevitably introducing noise to task-irrelevant circuits.
> > >
> > > In contrast, circuit-tuning performs a local structural update (sparse subgraph optimization). It primarily has two advantages:
> > > a) It identifies the *"intrinsic graph"* and leaves the majority of the computational pathways explicitly untouched, while simultaneously guaranteeing the performance of fine-tuning. This point is well-demonstrated in the experimental analysis in Section 4.5.
> > > b) It restricts the parameters required for the target task to a very small subgraph, thereby encouraging *the sparsity of the critical structure* relative to the specific task, and this kind of sparsity is what mechanistic interpretability encourages.
> > >
> > > Among the various variants of LoRA, a subset of methods focuses on mitigating the problem of catastrophic forgetting during fine-tuning. Representative works include **MiLoRA [1] and PiSSA [2]**. These methods perform continued training in a shared vector space, controlling the learning process by adding a regularization term on loss or applying specific optimization designs. MiLoRA and PiSSA use Singular Value Decomposition (SVD) components of the original parameters for LoRA initialization, with MiLoRA using minor components while PiSSA uses major components.
> > >
> > > In terms of computational cost, low-rank fine-tuning methods based on LoRA are lower than circuit-tuning. As shown in Table 2, the number of trainable parameters for circuit-tuning is *4.27 to 9.09 times* higher than that of LoRA (PiSSA/MiLoRA has the same trainable parameters as that of LoRA).
> > > On one hand, we believe this is because the goal of circuit-tuning is not originally designed to achieve parameter-efficient fine-tuning, but to study the internal changes of the model during fine-tuning. Therefore, it has not been optimized for parameter efficiency. In contrast, methods like MiLoRA and PiSSA still belong to optimization within a parameter subspace, making them more parameter-saving.
> > >
> > > However, the built-in pruning-then-fine-tuning approach of circuit-tuning gives it the characteristic of saving computational resources. Compared to full fine-tuning, it already saves *10.67 to 13.07 times* the parameters, making it efficient in practice, especially in mechanistic interpretability research.
> > > On the other hand, the trainable parameters introduced by circuit-tuning are *not* all significantly optimized during training. For example, as we analyzed in Finding 2 of Section 4.4, the function of the Object Identification Heads remains largely unchanged before and after fine-tuning. This suggests that circuit-tuning can learn to fine-tune selectively, meaning that for parameters in the circuit *corresponding to functions that are task-relevant but need to be shared before and after fine-tuning*, it smartly keeps them unaffected, thereby avoiding damage to general capabilities. This is also a point we find very interesting.
> > >
> > > Furthermore, **ReFT (Representation Fine-Tuning) [3]** is also an interpretability-based PEFT method, inspired by causal tracing. It achieves fine-tuning by learning interventions on activation values, i.e., it learns how to perform interventions that steer the model towards a target task. While our method also utilizes causal tracing, its goal is to locate task-relevant parameters by finding the circuit for the target task, and the final adjustment is still on the parameters.

---

> > > > ### Author Response · Authors · 2025-11-24
> > > > **Continue**
> > > >
> > > > **Intrinsic graph vs. Intrinsic dimension**
> > > >
> > > > LoRA-based methods are all based on the concept of intrinsic dimension. While the parameters for a task may not be decoupled from others and are perhaps distributed over the whole parameter space $\mathbb{R}^{D}$, prior works on intrinsic dimensionality used a reparameterization trick to separate the intrinsic dimensions $θ^d \in \mathbb{R}^{d}$ d out from the original parameter space. This kind of measurement aims to look for the upperbound of **the minimum description length (MDL) [4]** in terms of a task (the MDL principle encourages us to minimize the combined cost of encoding the model and the data given the model).
> > > >
> > > > Circuit-tuning is also derived from the idea of intrinsic dimension. While we aim to (i) find out the minimal subgraph $C*$ from a computational graph and (ii) interpret the learning process of a model based on the subgraph, there is a need to balance the minimum of the required parameters and the convenience for interpretable graph structure. Since we want to keep the original topological structure of the computational graph for further analyses, we directly delve into the original graph to inspect the redundancy, instead of adding extra structures like linear projections. Therefore, the minimal subgraph does not mean the minimum parameters in a strict sense.
> > > >
> > > > However, the functionalities of the nodes in a computational graph are thought to be decoupled from each other in the mechanistic study. **Braun et al. [5]** argued that one of the properties of a good decomposition of a model is the minimality that the decomposition should use as few components as possible to represent a model behavior of interest, which also follows the MDL principle. Therefore, circuit-tuning holds for the MDL principle in the sense of the minimality in mechanistic interpretability. Different from prior works, we do not require a strict estimation of the intrinsic dimensionality, but trying to take into account both the minimality of the graph and the feasibility for interpretability.
> > > >
> > > >
> > > > **Comparison with Model Pruning Methods**
> > > >
> > > > The parameter selection strategy in circuit-tuning is based on activation values and gradients, with causal tracing being the underlying idea.
> > > > Taking attribution patching as an example, this method measures the change in output using a metric $L$ by disrupting the activation value of a node $n$ in the model $M$, and this change is used as the node's importance score. This importance score can be estimated via a first-order Taylor expansion, specifically: $I \approx [n(x^{noise})-n(x^{clean})]^T\nabla_{n}L[M(x_{clean})] |_{n=n(x^{clean})}$.
> > > >
> > > > A similar idea also appears in the field of model pruning: *If the removal of a parameter has a large influence, then the model is sensitive to it and we should retain it*. For instance, for a parameter $w$, **Molchanov et al. [6]** estimate the difference in the loss function before and after $w$ is removed using a first-order Taylor expansion and use this as the sensitivity score for parameter $w$. Different from PEFT, model pruning aims to prune a model after training to reduce computational cost.
> > > >
> > > > In fact, **AdaLoRA [7]**, in order to address the issue that LoRA cannot adaptively select the rank $r$, also uses the sensitivity score $|w\nabla_{w}L|$ as a reference indicator for singular value selection, which is inspired from the idea of model pruning.
> > > > The difference is that AdaLoRA directly measures the importance of model parameters, while circuit-tuning calculates the importance of nodes (activations) or edges in the computational graph, since in most cases a model is viewed as a graph in the field of mechanistic interpretability. AdaLoRA uses the sensitivity score $|w\nabla_{w}L|$ to dynamically select different ranks for the adapters in different parts of the model, while circuit-tuning uses the edge contribution to decide which part of a model should be frozen, in order to isolate a circuit for the target task.
> > > >
> > > > We will consider including the above analysis content in the final version of the paper. Thank you again for your valuable feedback!
> > > >
> > > > ___
> > > > **Reference**
> > > >
> > > > [1] MiLoRA: Harnessing Minor Singular Components for Parameter-Efficient LLM Finetuning https://arxiv.org/pdf/2406.09044
> > > >
> > > > [2] PiSSA: Principal Singular Values and Singular Vectors Adaptation of Large Language Models https://arxiv.org/pdf/2404.02948
> > > >
> > > > [3] ReFT: Representation Finetuning for Language Models https://arxiv.org/pdf/2404.03592
> > > >
> > > > [4] Modeling by shortest data description https://doi.org/10.1016/0005-1098(78)90005-5
> > > >
> > > > [5] Interpretability in parameter space: Minimizing mechanistic description length with attribution-based parameter decomposition https://arxiv.org/pdf/2501.14926
> > > >
> > > > [6] Importance Estimation for Neural Network Pruning https://arxiv.org/pdf/1906.10771
> > > >
> > > > [7] Adalora: Adaptive budget allocation for parameter-efficient fine-tuning https://arxiv.org/pdf/2303.10512

---

> > > > > ### Comment · Reviewer_MVC6 · 2025-11-26
> > > > > **I meant a thorough experimental comparison with PEFT methods would be needed**
> > > > >
> > > > > Sorry, let me be more clear what kinds of comparisons I feel are needed:
> > > > > - To be interesting as a PEFT method, you'd need to convincingly demonstrate some direct practical advantage over existing methods; this would necessitate extensive experiments, and seems like to large a change to the current submission.
> > > > > - For the interpretability results/framing (the current focus), I think it's not clear there is sufficient novelty; the "new research paradigm" of developmental interpretability is discussed a lot, but most of the experiments are just looking at before and after (and not during) fine-tuning.  Perhaps a better understanding of related work could make me consider this work more novel than I currently do, but I'd be surprised.  On the other hand, I do think the work would still benefit from clearer positioning in the literature.

---

> > > > > > ### Author Response · Authors · 2025-11-27
> > > > > > **We decide to make it better.**
> > > > > >
> > > > > > Thank you for your advice. We decide to stop to revise our paper. From the very beginning of this study, we are targeted at interpretability, not PEFT, though circuit-tuning has some property like that. Next time, we'll carefully fix the words in our paper that are not rigorous, and conduct more experiments related to our topic.
> > > > > >
> > > > > > We've learned a lot from you. Thanks again and good luck!

---

### Official Review · Reviewer_d6Ta · 2025-10-31

**Soundness:** 3
**Presentation:** 4
**Contribution:** 3
**Rating:** 8
**Confidence:** 2

**Summary:**

This paper presents a novel fine-tuning method that, through the lens of intrinsic dimensionality, considers a pretrained model as a computational graph which is then iteratively pruned to a subgraph whose parameters are learnt for the specific task. The method is tested on a specially designed task that demonstrates its workability, in addition to other tasks.

**Strengths:**

1. Fine-tuning large models efficiently is pertinent to today's research landspace. This paper offers a fresh perspective on how that can be done based on several existing ideas upon which the proposed method is built, such as intrinsic dimensionality and circuit analysis. Packaging fine-tuning with mechanistic interpretability is a novel propostion, with promising initial insights that can guide further research.

2. A relatively simple but effective task is designed to validate the method, which also provides the reader with the opportunity to see the method in action and develop a better understanding of it.

3. The paper is well-written and easy to follow, even without much background knowledge in the area.

**Weaknesses:**

1. top-N is a task-relevant hyperparameter that is likely expensive to tune.

While limitations of the current study are discussed in the appendix, I think there are no major weaknesses at this stage.

**Questions:**

1. What is the primary purpose of this method, i.e. fine-tuning or dynamic interpretability?

2. What is meant by the 'rate of change in performance' on line 424? Is it simply the percentage of changed outputs before and after fine-tuning?

---

> ### Author Response · Authors · 2025-11-19
>
> Thanks a lot for your recognition of our work and your kind suggestions. I'll try my best to answer your concerns!
>
> **1. For weakness**
>
> **About Top-$N$**
>
> Top-$N$ refers to the number of edges in the circuit corresponding to the target task. In the experiments in Section 4, we conducted comparative experiments across multiple $N$ values (from 10 to 30k). We found that when $N$ exceeds a certain threshold, the performance improvement is very limited, as shown in Figure 5, where $N=1000$ serves as a knee-point.
>
> Based on this finding, when we experiment with larger models and more complex tasks in Section 5, we determine the value of $N$ using a method that incurs almost no time or computational resource cost before the formal experiments begin.
> Specifically, as described in Section 5.2, we first set a candidate set of $N$ values. For each $N$, we only need to perform circuit discovery on a single batch of training data to obtain the target task's base circuit. We then calculate the faithfulness and completeness of this "base circuit" (i.e. the circuit before fine-tuning). Finally, we select the number of edges corresponding to the circuit with larger faithfulness and smaller completeness as the value of $N$ for training.
>
> Note that even though the circuit may change during fine-tuning, based on our empirical findings, the circuits before and after fine-tuning share a large proportion of their structure. This means it is possible use the number of edges corresponding to the pre-fine-tuning circuit to determine the value of $N$ required for circuit-tuning. Therefore, the selection of $N$ is not that costly, a fact our experiments also verified.
>
> **2. For Questions**
>
> **(1) The primary purpose of circuit-tuning"**
>
> The primary motivation of circuit-tuning is the latter: to study the mechanistic interpretability of learning dynamics in a dynamic manner. We believe the most significant part is offering a new perspective to the research community, which is to actively apply interpretability-related techniques during training and study internal model structures like circuits from a dynamic view. This is what excites us the most~
>
> **(2) About "the rate of change in performance"**
>
> Let us formally define the rate of change in performance. Suppose our target task is $T_0$, and the tasks used for the evaluation of  general capabilities are $T_1, \dots, T_n$. If the model's performance on the test sets of $T_1, \dots, T_n$ (measured by accuracy or other metrics) before and after training are $a_1, \dots, a_n$ and $a'_1, \dots, a'_n$, respectively, then the rate of change in performance $\Delta a$ is defined as:
>
> $$\Delta a = \frac{1}{n} \sum_{i=1}^{n} \frac{a'_i - a_i}{a_i}$$
>
> In practice, for fairness, all evaluation results are the average of 10 runs with random seed.
>
> We hope this explanation would help you understand this metric.
>
> ---
> Thanks again for your time devoted in our work. Hope this response could further relieve your concerns!

---

### Official Review · Reviewer_DK8y · 2025-11-04

**Soundness:** 2
**Presentation:** 1
**Contribution:** 3
**Rating:** 2
**Confidence:** 4

**Summary:**

The authors propose to fine-tune a computational subgraph or a circuit instead of the entire model during fine-tuning and this helps them achieve slightly improved performance as compared to LORA fine-tuning. In case of subject-verb agreement the authors demonstrate to achieve better gains as compared to full fine-tuning. While analyzing the circuits at different iterations of fine-tuning, the authors also find the learning process to be upweighting and downweighting several edges in the circuits. They also find some edges being stronger than others and they get strengthened over the course of training. The similarity between the computational graphs before and after fine-tuning is around 20-25% (Fig. 11) which seems interesting.

**Strengths:**

* The direction analyzed by the authors seems quite interesting. I really liked their approach on using circuits to investigate fine-tuning.
* The results in section E.4.3 are quite interesting. However, it would strengthen the paper significantly if they organized properly in the main paper.

**Weaknesses:**

* I think the paper could significantly be improved in terms of writing (especially Sec. 4.4). It is hard to follow Sec. 4.4 and their circuit pruning algorithm. The current draft is focussed more on proposing an algorithm which could be used instead of fine-tuning to achieve improved results. However, I think the empirical evidence (Table-1) is not very convincing as the gains are quite minimal. Instead, if the authors would focus more on analyzing how the circuits change over the course of training (as in Sec. E.4.3), it can give more interesting insights.
* The number of parameters in table-1 for lora fine-tuning and circuit fine-tuning are different, where the circuit fine-tuning method uses a larger number of parameters, and thus these results are not fairly comparable.
* Several of the design choices in case of circuit pruning algorithm are unclear and it is also important to fairly compare different circuit pruning methods. This will help the authors justify their design choices specific to circuit pruning.

**Questions:**

It is not clear if the authors are doing node ablation also and how they are doing it. Could the authors provide more details on this? My understanding is that for tasks like subject-verb agreement the authors define and perform activation patching on either the node or the edge corresponding to the last token and then analyze the indirect effect of doing it. This gives them a metric. In case of other tasks the authors perform mean ablation where the mean is taken across all the samples and token positions for both edge and node pruning. However, there are some concerns  here and it would be great if the authors can clarify them:


Ideally a circuit should be defined for a single token prediction and a single sample as the computational graph for every token position as well as sample would be different. However, this would make the approach proposed in this paper quite intractable to do in practice. As a result there are several approximations done here. For instance ablating the nodes and edges by performing average over the indirect effects on all samples. This seems less principled by design although I understand that some previous papers also use this in the literature. It would be great to know how the authors think about this. I think my main criticism is about taking average over the token positions in case of mean ablation to determine the importance of a node/edge. Taking average over samples still makes sense, although I don’t find it to be completely principled. Can the authors also provide more details on why they chose specific choices for circuit pruning. How does their analysis compare with different circuit pruning methods? More ablations on this would be helpful.

---

> ### Author Response · Authors · 2025-11-19
>
> Thanks a lot for your kind suggestions. We'll try our best to answer your concerns!
>
> **1. For weakness**
>
> **(1) The structure and writing of our paper.**
>
> This is just what we worried about when writing this paper, and we apologize for any difficulty in your reading. This  issue has been a major focus of the revisions of this paper throughout the whole year of the submission process. We believe that:
> * It is necessary to focus more on learning dynamics. In Section 4.4, we merely enumerated the main findings and did not discuss them systematically in detail, instead placing most of the content in the Appendix. This was mainly due to the page limit. However, as you pointed out, the enhanced fine-tuning is an additional but non-primary contribution of circuit-tuning. Therefore, we will increase the detailed analysis regarding learning dynamics in the main body!
> * In previous versions of our paper, Section 5 did not take up much space. However, some reviewers felt that the discussion on how circuit-tuning mitigates the loss of general capabilities was insufficient. Thus we added more about it in this version, which unfortunately encroached upon the discussion of learning dynamics. Based on your suggestion, we guarantee that we could balance the relationship between Section 4.4 and Section 5 in the final version of the paper, especially by including more valuable content like the analysis in Appendix E.4.3.
> Incorporating the feedback from multiple reviewers, we have provided a new version of the paper to improve our previous paper structure and offer more experimental details. Thanks again for your advice!
>
> **(2) "The results in Table 1 are not fairly comparable"**
>
> This might be a little misunderstanding~ In Section 5, we conducted extensive hyperparameter searches for different methods. The results presented in Table 1 are the *best achievable results* we could find for each method.
>
> As discussed in Section 5.3, when we increase the number of tunable parameters for LoRA (by increasing LoRA's rank, alpha, and introducing more tunable components), its performance doesn't improve effectively. That is to say, even if LoRA had the exact same number of tunable parameters as circuit-tuning, its performance could not be better. This is because, during the inference time when the parameters of the LoRA adaptor are added to the base model, all parameters are changed. In contrast, circuit-tuning is designed to selectively fine-tune critical parameters, leaving task-irrelevant parameters unchanged, thus resulting in a higher performance ceiling than LoRA.
> Therefore, the experimental results in Table 1 ensure fairness, and we hope this clarifies your concerns~

---

> > ### Author Response · Authors · 2025-11-24
> > **Continue**
> >
> > **(3) Different circuit pruning methods**
> >
> > We have considered this when deploying circuit-tuning. In early experiments on the subject-verb disagreement task, we explored the most commonly used circuit discovery methods, including:
> >
> > * activation patching (patching nodes) [1]
> > * attribution patching (patching nodes) [2]
> > * path patching (edge attribution patching, EAP) [3]
> > * our modified EAP*
> >
> > Under the same experimental settings in Section 4.2, We only replaced the graph pruning strategy and repeatedly adjusted $N$ (the number of nodes or edges) to perform a fair comparative experiment, in order to investigate which method is best suited for circuit-tuning. We compared them and chose EAP for the following reasons:
> > 1. Empirical results show that EAP had the fastest convergence speed when measured by the logit difference. We observe that path patching (EAP) achieved higher logit difference than node attribution patching (0.937 v.s. 0.901), and the convergence speed is faster. We think it is because that compared to patching nodes, patching edges focuses more on the interactions between pairs of nodes. It can give an extremely zoomed in understanding of what's going on in the model and how parts compose together [2]. This also allows us to perform more detailed analyses on learning dynamics such as the one in Finding 5, on the strength of connections between nodes, which is more interesting than analyzing individual nodes.
> > 2. EAP is more efficient than standard activation patching. For circuit-tuning, the training efficiency would be slow if the time for graph pruning is too long. Standard activation patching is too slow, since it requires one forward pass and one backward pass for each node. Consequently, most of the time was wasted in the graph pruning stage, resulting in low GPU utilization. Therefore, we choose EAP because it manages to balance both performance and efficiency via only *one forward pass* for graph pruning for all edges, instead of running a forward pass for each node/edge.
> > 3. As for the modified EAP, we mention it in Section 4.2 and discuss it in details in Appendix E.4.5 (we will consider put it in the main body in the final version!). To be brief, we find that the original definition of the attribution score in EAP [3] fails to capture all the relevant edges in a task. We assume that there exists a situation where an important node is connected with many other nodes, but each edge is not that strong, as illustrated in Figure 12. Consequently that important node may fail to appear in the circuit.
> > Thus we modify the edge contribution of an edge $e: n_{i}^{u} \rightarrow n_{j}^{d}$ as below:
> > \begin{equation}
> >     c(e)^{\prime} = c(e) \cdot \sum_{k=1}^{N_{down}^{i}}c(n_{i}^{u} \rightarrow n_{k}^{d}) \cdot \sum_{k=1}^{N_{up}^{j}} c(n_{k}^{u} \rightarrow n_{j}^{d})
> > \end{equation}
> > where $c(e)$ is the original attribution score in EAP, $N_{down}^{i}$ is the number of the downstream nodes of $n_{i}^{u}$, $N_{up}^{j}$ is the number of the upstream nodes of $n_{i}^{u}$.
> > The revision considers the contributions from all the edges connected to the upstream node and the downstream node. As shown in Figure 13 and Table 4, our modification achieves better performance than original EAP on the subject-verb disagreement task, and even bringing down the computation cost.
> >
> > We'll add more quantitative results about it to the paper.

---

> > > ### Author Response · Authors · 2025-11-24
> > > **Continue**
> > >
> > > **2. For Questions**
> > >
> > > **About mean ablation**
> > >
> > > In our view, mean ablation serves as a random activation to replace the corrupted activation in standard causal tracing. We believe that *no matter how the corrupted activation is calculated, its ultimate goal is to find noise that disrupts the original activation*.
> > >
> > > Taking attribution patching as an example, this method measures the change in output using a metric $L$ by disrupting the activation value of a node $n$ in the model $M$, and this change is used as the node's importance score. This importance score can be estimated via a first-order Taylor expansion, specifically: $I \approx [n(x^{noise})-n(x^{clean})]^T\nabla_{n}L[M(x_{clean})] |_{n=n(x^{clean})}$. In practice, we often use the absolute value of the importance score, and $n(x^{noise})$ is often estimated via mean/zero ablation. Next, let’s discuss about the ablation method.
> > >
> > > Let’s consider a node $n$ (e.g. an MLP output in a layer) with an activation value of $n(x)=(a_1, a_2, …, a_n)$. According to the discussions of “features” in a neural network [4], each neuron corresponds to one or multiple (when superposition exists) features in the real world. For  convenience, let the $m$-th neuron $(1 <= m <= n)$ to be the neuron of  interest, i.e., it corresponds to the features for the target task.
> > >
> > > The best way to perform causal tracing is to find a corrupted prompt $x’$ that is formally consistent but semantically opposite to the clean prompt $x$. This ensures that the difference between the clean and corrupted activations at the target position (i.e. the position used to predict the key word, such as the END token in the subject-verb disagreement task) is focused specifically on features relevant to the target task. In this case, the ideal corrupted activation of the node $n$ should be $n(x’)=(a_1, a_2, …, a’_m, …, a_n)$, such that $n(x)-n(x')=(0, 0, …, a_m-a’_m, ..., 0)$.
> > >
> > > As you mentioned, previous work also adopts mean/zero ablation to compute the corrupted activation. In our experiment, we compute the average activation across all tokens in all samples. In our view,  this is equivalent to randomly sampling an activation that approximates Gaussian noise at the current node's position from the training data. Let’s use $n(x’’)=(a’_1, a’_2, …, a’_m, …, a’_n)$ to denote the corrupted activation value in this case. The activation value $a’_i$ of each neuron measures the average level of the activation at that neuron. Here, the difference $n(x)-n(x’)=(a_1-a’_1, a_2-a’_2, …, a_m-a’_m, ..., a_n-a’_n)$.
> > > If the node $n$ is *not* important for the target task, then the activation of any neuron including the $m$-th neuron would not be significant. Then $a_i-a’_i (1<=i<=n)$ is supposed to be similar. On the contrary, if the node $n$ is important for the target task, then the $m$-th neuron would  fire significantly. Then $a_m-a’_m$ is supposed to be larger than $a_i-a’_i (1<=i<=n, i \neq m)$, and thus emphasizing the significance of this node.
> > >
> > > Therefore, mean ablation over all tokens of all samples in a batch is theoretically effective, though it may not be as good as the corrupted activation derived from a contrastive prompt.
> > > In fact, we also considered this issue in our experiment. To ensure that our mean ablation works as expected, we computed the faithfulness and completeness of the circuit corresponding to the target task in Section 4.5 to ensure that our method could reliably find the task-relevant circuit. The experimental results confirm this point.
> > >
> > > Hope our analysis could further clear your concerns!
> > >
> > > ---
> > > Thanks again for your time on our work. We are willing to reorganize the content and structure of the paper according to your suggestions to improve the reading experience.
> > >
> > > We believe circuit-tuning is promising and really hope that you could re-evaluate it. Looking forward to your reply !!
> > >
> > > **Reference**
> > >
> > > [1] Towards Automated Circuit Discovery for Mechanistic Interpretability https://arxiv.org/abs/2304.14997
> > >
> > > [2] Attribution patching: Activation patching at industrial scale https://www.neelnanda.io/mechanistic-interpretability/attribution-patching
> > >
> > > [3] Attribution Patching Outperforms Automated Circuit Discovery https://arxiv.org/abs/2310.10348
> > >
> > > [4] Toy Models of Superposition https://transformer-circuits.pub/2022/toy_model/index.html

---

### Official Review · Reviewer_iyxq · 2025-11-06

**Soundness:** 3
**Presentation:** 3
**Contribution:** 3
**Rating:** 6
**Confidence:** 4

**Summary:**

This paper proposes circuit-tuning, wherein a process like alternative optimization is setup to localize the effects of fine-tuning on a task. Specifically, one isolates the specific set of parameters that, when updated, are likely to be sufficient to capture the effect of full fine-tuning. Fine-tuning is then performed for a few steps, and the set of parameters allowed to updated is recalculated. This process then keeps repeating in a periodic manner. While there's prior literature with this flavor of methodology for adding or removing capabilities from a model, I think the precise results achieved by this paper are really cool.

**Strengths:**

Section 4, i.e., the main findings, is really thorough and well done (even if a lot of back and forth between appendix and main paper is needed). I can easily see this analysis promoting further work down the line for using fine-tuning to do hypothesis testing for circuits.

**Weaknesses:**

- Relation to prior work: My main apprehension with the paper is how it's contextualized with respect to the broader literature. Specifically, similar approaches as this paper were proposed in work like [1] to iteratively add a task to a model by identifying a parameter mask that maximally aids learning of a task without interfering with other learned tasks. The goal here was continual learning and the precise domain / model class is different, but subsequent papers from this group generalizes these assumptions. I strongly recommend the authors look into these papers and properly contextualize their work. On a related note, I'll also highlight the paper by Panigrahi et al. [2], which demonstrates the ability to isolate effects of fine-tuning (in MLMs) to subnetworks with extremely small set of parameters.

- Narrative of explaining fine-tuning: At times the authors use statements that are suggestive of how their analysis methodology offers an explanation for how fine-tuning changes a model's behavior. I think these statements warrant more tempering: the explanation is more like a hypothesis that's been derived out of use of a specific method. As an analogy, consider how methods like LoRA can, at times, be used on a specific layer to capture the effect of full fine-tuning. Does that mean full fine-tuning has solely a low-rank impact on a specific layer in the model? Probably not. Relatedly, I think the paper title is currently over-claiming the findings (which are cool by themselves and do not need this flashiness). Specifically, saying fine-tuning is subgraph search seems unnecessary to me; one could simply suggest restricting fine-tuning to subnetworks elicits interesting phenomenology of how fine-tuning might work.

Minor:
- Line 150 seems to be missing a word around the phrase "refer to those __ words"

[1] https://arxiv.org/abs/2006.14769
[2] https://arxiv.org/abs/2302.06600

**Questions:**

The measure "average tunable parameter ratio" didn't quite make sense to me. Can the authors rephrase what this measure is, how it's formally defined, and what is the conclusion shown in Figure 5a accordingly? I expect the claim is not that 50% of the model parameters have been changed?

---

> ### Author Response · Authors · 2025-11-19
>
> Thanks a lot for your time on our paper and kind suggestions for our work. We are happy to know that you are interested in our work! We'll try our best to answer your concerns.
>
> **1. For weakness**
>
> **(1) Relation to prior work**
>
> Thanks for your advice. During the development of circuit-tuning, we did realize that our work has something to do with continual learning and model pruning. Some work also treats a neural network as a computational graph and tries to find out the “empty parts” that could be utilized for new tasks. On the technical level, gradient-based methods are used in the areas of both continual learning and mechanistic interpretability, for the localization of the important components in a model for a specific task. As for the papers you recommended to me (*SupSup & skill localization*), I think they are really cool and we do share a similar idea! That is to dynamically separate a subgraph for each task. Bringing this idea into the training of a model could encourage the sparsity of task-specific model components, and further relieve problems like catastrophic forgetting.
>
> Due to page limitations during the submission process, the relevant content was not included in the main body. However, we do believe it is essential to also state the relevant content from fields such as continual learning and compare it with circuit-tuning! If our paper could be accepted, we will surely mention and discuss the content you recommended, along with all other related content we have researched in the final version of our paper.
>
> **(2) Narrative of explaining fine-tuning**
>
> It is true that the title of our paper is kind of misleading. Generally speaking, normal fine-tuning (full fine-tuning) is definitely not "subgraph search". What we mean is that with the help of circuit-tuning, fine-tuning could exhibit behavior similar to subgraph search, and the introduced sparsity could help us learn more about the development of a circuit for a task. We will adjust the title to more precisely convey the core idea of our work, e,g, *"Circuit-tuning: A Mechanistic Approach for Identifying Parameter Redundancy and Fine-tuning Neural Networks"*. Anyway, thanks for pointing this out~
>
> **(3) For the typo**
>
> We will fix it !
>
> **2. For Questions**
>
> The "average tunable parameter ratio" measures the proportion of the circuit in the entire computational graph. Here is the definition of it:
>
> * Suppose the number of the parameters in a model is $N_p$ (bytes). During circuit-tuning, we iteratively find a subgraph and fine-tune the parameters in it. At each iteration $i$, if the number of the parameters in the subgraph is $n_i$, then the ratio of parameters involved for fine-tuning is $\frac{n_i}{N_p}$. If the total number of iterations throughout the fine-tuning process is $M$, then the average parameter ratio is $\sum_{i=1}^{M} \frac{n_{i}}{N_{p}}$.
>
> From Figure 5(a), it can be seen that the average tunable parameter ratio is 0.45 when the number of edges $N$ surpasses 5000. Note that the ratio of parameters that have been changed during fine-tuning is not necessary 0.45, but *less than* it. Among the 45% parameters, a large proportion remain essentially unchanged during the fine-tuning process, such as the parameters responsible for subject identification mentioned in Section 4.4. These parameters are important for the task, but since the function they perform remains the same before and after fine-tuning, their values are almost unchanged (the gradients on them are quite small).
>
> Furthermore, the granularity of the nodes in the computation graph also affects this value. In the subject-verb disagreement task, we treated an entire MLP parameter matrix as a single node, which inevitably includes many redundant parameters. If we were to split the MLP into smaller nodes, the value of the average tunable parameter ratio would decrease further.
>
> Therefore, the "average tunable parameter ratio" actually describes the proportion of the critical subgraph relative to the entire computation graph. It is different from the one used in parameter-efficient fine-tuning. From this view, the name of this metric is also kind of misleading. Perhaps we should call it the *"average subgraph ratio"*? We will discuss it in detail in the final version of our paper.
>
> ---
> Finally, we hope our response could further relieve all of your concerns. Thank you again for your time on our work !!!

---

### Note · Authors · 2025-11-27

I have read and agree with the venue's withdrawal policy on behalf of myself and my co-authors.